# Molecular Insights into the Diagnosis of Anaplastic Large Cell Lymphoma: Beyond Morphology and Immunophenotype

**DOI:** 10.3390/ijms26125871

**Published:** 2025-06-19

**Authors:** Jesús Frutos Díaz-Alejo, Iván Prieto-Potín, Rebeca Manso, Marta Rodríguez, Marcos Rebollo-González, Francisco Javier Díaz de la Pinta, Miriam Morales-Gallego, Socorro María Rodríguez-Pinilla, Arantza Onaindia

**Affiliations:** 1Pathology Department, Instituto de Investigación Sanitaria-Fundación Jiménez Díaz University Hospital, Universidad Autónoma de Madrid (IIS-FJD, UAM), 28040 Madrid, Spain; jesus.frutos@quironsalud.es (J.F.D.-A.); ivan.prieto@quironsalud.es (I.P.-P.); marta.rodriguezm@iis-fjd.es (M.R.); marcos.rebollo@iis-fjd.es (M.R.-G.); fjavier.diazp@quironsalud.es (F.J.D.d.l.P.); miriam.moralesg@iis-fjd.es (M.M.-G.); smrodriguez@quironsalud.es (S.M.R.-P.); 2 Pathology Department, Osakidetza Basque Health Service, Araba University Hospital, 01070 Vitoria-Gasteiz, Spain; arantza.onaindiaperez@osakidetza.eus; 3Oncohaematology Research Group, Bioaraba Health Research Institute, 01070 Vitoria-Gasteiz, Spain

**Keywords:** Anaplastic Large Cell Lymphoma, ALK, DUSP22, TP63, STAT3, BIA-ALCL, pcALCL

## Abstract

Anaplastic Large Cell Lymphoma (ALCL) represents a diverse group of mature T-Cell Lymphomas unified by strong CD30 expression but with different molecular and clinical subtypes. This review summarizes recent molecular advances in ALCL, highlighting key discoveries that have refined its classification, diagnosis, and therapeutic strategies. ALCL comprises four major entities: systemic ALK-positive ALCL, systemic ALK-negative ALCL, Breast Implant-Associated ALCL (BIA-ALCL), and primary cutaneous ALCL. Each subtype exhibits unique phenotypes, along with cytogenetic and molecular alterations that affect clinical outcomes. Nevertheless, different oncogenic mechanisms mediate STAT3 activation. In ALK-positive ALCL, ALK fusion proteins drive oncogenesis via constitutive activation of STAT3 and other signaling pathways. ALK-negative ALCL comprises heterogeneous genetic subtypes, in which JAK/STAT3 pathway alterations and novel gene fusions are gaining recognition as potential therapeutic targets. This review emphasizes the need for integrative molecular diagnostics to improve stratification of ALCL subtypes and targeted treatment approaches. Future research should focus on elucidating the biological mechanisms underlying these alterations and on translating molecular insights into clinical practice.

## 1. Introduction

Anaplastic Large Cell Lymphoma (ALCL) was first introduced in the Kiel classification of 1988 [1] and in the Revised European American Lymphoma (REAL) classification of 1994 [2]. In 1997, the ALK1 (Anaplastic Lymphoma Kinase 1) monoclonal antibody suitable for FFPE sections was developed [3,4]. In the late 1990s, new *ALK*-related translocations were identified [5,6,7,8] and in 2001 ALCL was finally included in the World Health Organization (WHO) classification [9]. Six years later, ALK-negative Anaplastic Large Cell Lymphoma (ALK- ALCL) was included as a provisional entity in the WHO classification [10]. Subsequent genetic studies resulted in restriction of the diagnosis to cases of T-cell or null lineage. More recently, in 2016, ALK- ALCL was included as a definitive entity and Breast Implant-Associated ALCL (BIA-ALCL) entered as a provisional entity in the WHO classification [11], finally being included as a definitive entity in the provisional 5th revision of the WHO classification (WHO-HAEM5) [12] and the most recent version of the International Consensus Classification (ICC) [13].

The definition of ALCL has evolved since its original description in 1985 by Stein et al. [14]. ALCLs are now defined as mature T-Cell Lymphomas characterized by pleomorphic tumor cells with uniform strong expression of CD30, often defective expression of T-lineage markers and a tendency to grow cohesively and invade lymph node sinuses [15].

### 1.1. Mechanisms Leading to CD30 Expression

CD30 is the main marker for classifying a case as ALCL, but several other T-Cell Lymphomas can also express this protein [16,17]. In ALCLs, CD30 expression is intense, diffuse and observed in most neoplastic cells (>75%). Staining is frequently found in the membrane, cytoplasm and Golgi [18]. In contrast, the staining is less intense in other T-Cell Lymphoma subtypes, and considerably fewer neoplastic cells are positive for this marker. Table 1 summarizes the most important studies that have documented the expression of this marker in various T-cell malignancies. Other types of non-hematological malignancies may also show CD30 expression [19,20].

CD30 is a member of the tumor necrosis factor receptor superfamily and plays a crucial role in activating various oncogenic pathways, such as the induction of pro-survival signals and cell proliferation [15]. Overexpression of CD30 generates abnormal signals that lead to chromosomal instability [35]. Signaling originates through contact of CD30 with CD30L/CD153, facilitated by intracellular adapter proteins [36] such as TRAF isoforms [37]. These signal transducers activate canonical (regulated by TAK1 [38]) and non-canonical forms of the NF-κB pathway [39,40]. Additionally, aberrant signaling may be transferred to the inner cell via trogocytosis [16]. CD30 overexpression may be caused by various factors: hypomethylation of its promoter [41] and/or overexpression regulated by several transcription factors (JunB, IRF4, STAT3 or BATF3) forming the “superenhancer” transcriptional complex [42,43,44,45], which also activates different cellular regulation pathways, such as proliferation, cell cycle and apoptosis [46]. Gains of 1p6.22 locus [47], trogocytosis or infection by Human T-Lymphotropic Virus 1 (HTLV-1) have been identified as other causes of CD30 overexpression [16,48,49] (Figure 1).

### 1.2. ALCL Classification and Subtypes

WHO-HAEM5 [12] and 2022 ICC [13] currently recognize four entities within the ALCL family, as described below:Systemic ALK-positive Anaplastic Large Cell Lymphoma (**ALK+ ALCL**).Systemic ALK-negative Anaplastic Large Cell Lymphoma (**ALK- ALCL**).Breast Implant-Associated Anaplastic Large Cell Lymphoma (**BIA-ALCL**).Primary cutaneous Anaplastic Large Cell Lymphoma (**pcALCL**).

Systemic ALCL (sALCL) accounts for 3% of non-Hodgkin’s lymphomas (NHLs) in adults [50] and 10–15% of those in children [51,52] non-Hodgkin’s lymphoma (NHL) cases. ALK+ ALCL comprises approximately 3–7% of NHL in adults, followed by ALK- ALCL (1–2%). BIA-ALCL makes up 0.1–0.3% of them [53], although it accounts for a much higher percentage in the context of implant-associated lymphomas (80–90%). ALCLs of all types represent approximately 10–15% of Peripheral T-Cell Lymphomas (PTCLs), distributed between ALK+ ALCL (5–7%) and ALK- ALCL (4–8%) [53,54]. pcALCL comprises approximately 20–25% of Cutaneous T-Cell Lymphomas (CTCLs) and approximately 20% of all Cutaneous Lymphomas [55,56].

ALK+ ALCLs usually have good long-term survival rates in adults (approximately 70%) [57,58] and in children (10-year Overall Survival [OS] of 90%) [59]. On the other hand, ALK- ALCLs have worse outcomes, with a 5-year OS of 32% [27,54]. However, the prognosis varies among the subgroups. It is estimated that pcALCLs have an 80–95% 5-year OS, except for some rare aggressive cases with systemic involvement, whose outcomes are worse [55,60].

### 1.3. ALCL Etiopathogenesis

The most widely accepted theory about the etiopathogenesis of ALCL is that it is initiated in early thymocytes, before TCR-β rearrangement, and requires an initial transient burst of TCR signaling to initiate lymphomagenesis. This hypothesis is supported by the fact that some ALCL cells overexpress TARC (Thymus and Activation-Regulated Chemokine) or acquire thymic progenitor-like features [61,62,63].

Genome-wide DNA methylation profiling studies revealed that ALK+ and ALK- ALCL share methylation characteristics, such as modifications in TCR and CTLA-4. The general methylation status of most ALCL cases is similar to that presented by immature thymocytes; these results support the aforementioned theory [64]. Specifically, ALK+ ALCL methylation patterns are comparable to those of Early Thymocyte Progenitors (ETPs) and other mature T-cell subtypes (memory, regulatory, activated and helper T-cells). In the case of ALK- ALCL, neoplastic cells may derive from these mature T-cell and double-positive CD4+ CD8+ thymic precursors [65]. Finally, Iyer et al. studied the clonotype pattern of different T-Cell Lymphomas and observed that ALCL cells show low frequency of TCRβ and TCRγ DNA, reinforcing the idea of these neoplastic cells coming from a thymic precursor immature cell [66] (Figure 2). Nevertheless, the origin of ALCL remains unknown.

This review covers the main morphological, immunophenotypical, molecular and clinical features of each ALCL subtype, focusing on the new molecular findings of several studies which could help refine the classification and diagnosis of these tumors. These new approaches are required in order to improve the clinical management and outcomes of patients.

## 2. Systemic ALK-Positive Anaplastic Large Cell Lymphoma

### 2.1. General and Clinical Aspects

ALK+ ALCL is, by definition, a CD30-positive mature T-Cell Lymphoma (TCL) with aberrant expression of the ALK protein due to rearrangements of the *ALK* gene [12]. This group accounts for 50% of sALCL cases [15], which have a higher incidence in males (male:female ratio of 1.5) [67,68]. The mean age of patients is 34 years [54,68], although this varies considerably depending on patient demographics [67]. The incidence is higher in black than white people, but a lower incidence has been noted in native American and Asian people. The 5-year OS rate also varies by ethnicity: 49.7% in white people, 37.7% in black people, 42.8% in Asian people and 35.8% in native Americans [67].

Seventy-five percent of patients show advanced-stage disease at presentation and refer systemic symptoms [67,68,69]. Most cases (90%) present with lymph node involvement (LNI), and frequently with extranodal involvement (60%), mainly in skin, bone, soft tissue, lung [67,68,69] and bone marrow (BM) (10–14% of cases). Affectation of the Central Nervous System (CNS) [70,71] is rare.

### 2.2. Morphology and Immunophenotype

Morphologically, classic ALK+ ALCL usually presents diffuse/cohesive growth patterns in up to 60% of cases, commonly involving lymph node sinuses, characterized by large, pleomorphic lymphoma cells [72,73,74,75,76], although different patterns like paracortical and sinusoidal may be seen. The involvement pattern may be focal and subtle.

There are several morphological variants, such as lymphohistiocytic, small cell, Hodgkin’s-like, hypocellular, neutrophil-rich, sarcomatoid, and composite patterns [11,12,15,27]. The majority of ALK+ ALCL cases show at least some characteristic pleomorphic cells with abundant cytoplasm and horseshoe-shaped (“hallmark”), wreath-like or multiple nuclei, surrounded by an eosinophilic region. Sometimes, pseudoinclusions within the nucleus can be observed in the so-called “doughnut” cells, and rare mitotic activity can be observed. Despite the previously listed morphological variants, these hallmark cells may generally be found in all cases. BM forms of infiltration vary from extensive involvement by tumor cells to only a few scattered ones.

CD30 expression is diffuse. However, morphological and CD30 expression patterns may change between biopsies taken at different times during the course of the disease [77], as well as differing among morphological variants. For example, the small-cell variant is weakly ALK positive and presents strong CD30 expression in only 5–10% of cells, in contrast to the classic ALK+ ALCL pattern [15,69,75,76,78] (Figure 3).

The loss of several pan-T-cell markers, as well as a general lack of TCR expression [79], are also important immunophenotyping criteria [12,15]. The most frequently expressed T-cell markers are CD2 and CD4 (25–40% of cases), as well as CD5 (30% of cases), CD43 (40%), cytotoxic markers (TIA1, granzyme B and perforin in 75–90% of cases), CD25, Clusterin and BCL6. Expression of epithelial markers (EMA and CK, 80% of cases), myeloid antigens (CD13 and CD33), CD56 (10%) and stem cell transcriptional factors (including SOX2) [27,68,80,81] makes the differential diagnosis with many other necessary entities. pSTAT3 is expressed in most cases (75%). The extent of PD-L1 expression is notably higher in this subgroup than in other PTCLs, and is also associated with pSTAT3 expression, which makes PD-L1 a potential therapeutic target for ALK+ ALCL [82].

### 2.3. Cytogenetic Alterations

As previously mentioned, the immunophenotype of ALK+ ALCL is determined by the detection of ALK protein expression using a monoclonal antibody developed by Pulford in 1997 [4] that is immunohistochemically highly sensitive and specific. This phenomenon is strongly correlated with the presence of *ALK* gene rearrangements, which are detected by molecular study methods such as fluorescence in situ hybridization (FISH). These structural alterations fuse the 3’ portion of the ALK gene on chromosome 2p23 with the 5’ portion of a partner gene that provides the promoter, leading to constitutive expression of the chimeric oncoprotein and strong, constitutive activation of the kinase function [83,84]. Critical translocations of the *ALK* gene have been identified; that is why the combination of two techniques, immunohistochemistry (IHC) and FISH, is considered the gold standard for diagnosing ALK+ ALCL. Staining patterns differ depending on the associated *ALK* fusion partner (Table 2). Morphological variants may show contrasting ALK staining. For example, the larger cells in the sample show the classic ALK expression and pattern, while the small cells exhibit a nucleus-restricted pattern. This variability complicates the evaluation and diagnosis of patients [15,69,75,78].

Salaverria et al. identified chromosomal imbalances in 58% of their ALK+ ALCL cohort [85]. Gains of 17p/17q24, and losses of 4q13-q21/11q14 were the most frequently observed abnormalities. 4q13 loss has also been described in Multiple Myeloma and identified as tumorigenic due to the suppression of Platelet Factor 4, which may be a potential tumor-suppressor gene [86]. 11q losses are also present in variants of Burkitt Lymphoma and High-Grade B-Cell Lymphoma, Not Otherwise Specified [87]. None of the other chromosomal alterations had been described in any class of lymphoma type. Several genes of interest, located in the following regions, may be involved in the pathogenesis: RAB38 (11q14), *CXCL1*, *CXCL2*, *CXCL3*, *CXCL5* (4q13), *CXCL9*, *CXCL10*, *CXCL11*, *CXCL13*, *MAPK10* (4q21), *TP53* (mutated in 11% of ALK+ ALCL cases) [88,89], *PIK3R5*, *PIK3R6*, *MAP2K3*, *MAP2K4* and *USP22* (17p).

Common chromosomal alterations have been described in ALK+ and ALK- ALCLs, such as gains of 7p and 6q and losses of 13q [85].

**Table 2 ijms-26-05871-t002:** Summary of the main described partners involved in rearrangements with the ALK gene.

Chromosomal Translocation	ALK Partner	Partner Gene Function	% of Cases	Expression Pattern
t(2;5)(p23;q35)	*NPM1* [5,90,91,92]	Nuclear protein that shuttles between the nucleus and the cytoplasm	80	Nuclear and cytoplasmic
t(1;2)(q25;p23)	*TPM3* [92,93]	Cytoskeletal protein	13	Cytoplasmic
Inv(2)(p23q53)	*ATIC* [92,93]	Purine biosynthesis pathway	1	Cytoplasmic
t(2;3)(p23;q21) [92,93]	*TFG Xlong* *TFG long* *TFG short*	Associated with ER and microtubules	<1	Cytoplasmic
t(2;17)(p23;q23)	*CLTC* [92,93]	Component of the cytoplasmic face of intracellular organelles	<1	Cytoplasmic
t(2;X)(p23;q11.12)	*MSN* [92,93]	Submembranous cytoskeleton	<1	Cytoplasmic
t(2;19)(p23;p13.1)	*TPM4* [92,93]	Cytoskeletal protein	<1	Cytoplasmic
t(2;22)(p23;q11.2)	*MYH9* [92,93]	Cytoskeletal (major contractile protein)	<1	Cytoplasmic
t(2;9)(p23;q33–34)	*TRAF1* [92,93]	TNF signaling, signaling adaptor	<1	Cytoplasmic
t(2;11)(2p23;11q12.3)	*EEF1G* [92,93]	Translation elongation factor activity; subunit of the elongation factor-1	<1	Cytoplasmic
t(2;17)(p23;q25)	*RNF213/ALO17* [92,93]	E3 ligase	<1	Cytoplasmic
t(2;5)(p23;q35)	*SQSTM1* [94]	Regulates activation of the NF-kB signaling pathway	<1	Cytoplasmic
t(2;11)(p23;p13)	*CAPRIN1* [94]	ATP binding, scaffold activity and signaling adaptor	<1	Cytoplasmic

### 2.4. Molecular Alterations

Constitutively activated *ALK* fusion proteins promote tumorigenesis by activating diverse signal transduction pathways, including the PLC-γ [95], PI3K/AKT/mTOR [96,97], Cdc42/Rac1 [98], JNK [99], MEK/ERK, JAK/STAT3 [100,101] and STAT5 [102] pathways. Their activation promotes oncogenesis by enhancing cell survival, inhibiting apoptosis, promoting tumor dissemination and immune surveillance evasion mechanisms [101]. Furthermore, *NPM1::ALK* has been reported to control T-cell identity by epigenetic silencing of many T-cell-associated antigens (TCR-related signaling molecules) [83,103], such as DNMT1, CD3ε, ZAP70, LAT and SLP76, through STAT3-mediated regulation of gene transcription and/or epigenetic silencing [103], However, 74–90% of sALCL cases (ALK+ and ALK- ALCLs) feature clonal TCR rearrangements [79].

Despite its complex cellular signaling profile, STAT3 has a key central role arising from its regulation of the various target genes involved in the cell cycle, apoptosis, immune response, angiogenesis and metabolism [97,104,105,106,107]. In this context, STAT3 allows the ALCL cell to mimic progrowth signals (mainly via the IL2–STAT5 axis) [108,109]. The development of Gene Expression Profiling (GEP) technology has enabled the targets directly activated by STAT3 to be detected. One of the most relevant of these is *C/EBPβ*, an intronless gene involved in various cellular processes, including differentiation, proliferation, inflammatory response and metabolism [110]. This gene promotes the transcription of *BCL2A1* and *DDX21*, as well as myeloid antigens (CD13, CD33 and Clusterin) [105,111,112].

Another key target of *STAT3* is the Interferon Regulatory Factor 4 (*IRF4*) gene. Studies have shown that *IRF4* is essential for ALCL cell survival [113] which is mainly ensured by activating *MYC* transcription, regardless of the presence or absence of alterations of this gene. The IRF4–MYC axis also plays an important role in the pathogenesis of this disease through the upregulation of various genes, such as *BATF3* [114] and pathways that promote cell proliferation and survival (Figure 4).

A novel *STAT3* mutation was recently described in murine models that affects various hematological malignancies, including ALK+ ALCL. *SBNO2* proved to be one of the main transcriptional targets of STAT3^Y640F^. This newly described mutation of *STAT3* and its downstream activation of *SBNO2* could be a potential therapeutical target [115]. Recurrent somatic mutation in FAT family genes and *RUNX1T1* are also known to induce changes in morphology, growth, migration and treatment resistance in ALK+ ALCL cell lines, and to be associated with STAT1 and STAT3 overexpression [116].

Mutations in pediatric ALK+ ALCL patients with biological implication have recently been described in some genes [52], such as *TP53*, *MDM4*, *JUNB*, *TET1*, *KMT2B*, *KMT2A*, *KMT2C* and *KMT2E*. TET2 epigenetic modifications in the thioredoxin-interacting protein gene (*TXNIP*) in ALCL cells (ALK+ and ALK−) could play a crucial role in their cell cycle [117].

### 2.5. Role of Non-Coding RNAs in ALK+ ALCL

The main types of non-coding RNAs are microRNAs (miRNAs), long non-coding RNAs (lncRNAs), circular RNAs (circRNAs), piwi-interacting RNAs (piRNAs), small interfering RNAs (siRNAs), vault RNAs (vtRNAs) and small nucleolar RNAs (snoRNAs). All of them carry out key physiological functions, such as modification of chromatin, mRNA processing and ribosomal RNA maturation, among others. Nevertheless, only a few of these types in this tumor subclass have been studied (Table 3).

miRNAs play a crucial role regulating protumor signals in ALK+ ALCL. One of the first studies on this topic was published in 2010 by Merkel et al. [118], who described the upregulation of miR-17–92 cluster members in ALK+ ALCL in cell lines, transgenic mouse models and primary tumor tissues [118], which activates STAT3 [119]. They also discovered that miR-101 was also found to be downregulated in ALK+ ALCL, causing tumor-cell proliferation due to the activation of mTOR pathway [120,121]. miR-155 is another important downregulated miRNA. It favors Th2 differentiation and low levels of IFN-γ, thereby creating an immunosuppressive environment [122].

The following year, Matsuyama et al. [123] demonstrated the activation of the NPM-ALK−STAT3 axis by miR-135b, which causes T-cell polarization through IL-17 secretion, activating the transcription of *STAT6* and *GATA3*, which also leads to tumoral immunosuppression. miR-150 is another miRNA known potentially to have antitumor properties, and which is therefore downregulated by *NPM1::ALK* chimeric protein [124].

A novel marker for the early identification of high-risk pediatric patients, miR-146a-5p, has recently been described [125]. This microRNA was detected by small-RNA sequencing in plasma samples from 20 patients. The authors suggest that this marker could promote macrophage infiltration and polarization towards the M2 phenotype, increasing tumor aggressiveness and dissemination capacity.

Regarding the second type of non-coding RNA, lncRNAs, overexpression of LINC01013 was described after lncRNA-microarray analysis [126]. In vitro assays (KARPAS-299 ALK+ ALCL cells) subsequently demonstrated that LINC01013 depletion reduces invasive properties in this model due to the induction of Snail, which has been described as one of the major activators of the Epithelial–Mesenchymal Transition (EMT) [127,128].

Very little is known about circRNA in ALCL. Babin et al. noticed the formation of circRNAs that include the breakpoint sequence of *NPM1::ALK* [129] in murine models.

Finally, snoRNAs mainly act by modifying ribosomal RNAs, and are known to be generally downregulated in cancer cells [130], even though a fraction of them serve as a sign to characterize ALCL through Reverse Transcription Quantitative PCR (RT-qPCR) in a PTCL cohort [131]. U3 snoRNA expression appears to be specific to ALK+ ALCL, thereby making it possible to distinguish ALK+ and ALK- ALCL cases [131].

### 2.6. Prognosis and Prediction

There are remarkable molecular and morphological events that, if present, are associated with a worse prognosis. The first of these is related to CNS involvement at initial presentation associated with lymphohistiocytic variant morphology and leukemic presentation with myeloid antigens, which is unusual in this disease [132,133,134,135,136]. A recent discovery is that TP53 deletions are associated with poor survival in adult ALK+ ALCL patients [137].

Some morphological features have also been associated with poorer outcomes in pediatric patients, such as small-cell and lymphohistiocytic patterns, which are associated with relapsed or refractory diseases in these patients [59,138,139]. *NPM1::ALK* transcripts detected by RT-qPCR in peripheral blood are a marker of poor clinical progression in these young patients [140,141]. Finally, pediatric ALK+ ALCL cases can be divided into two groups based on their ALK expression levels, suggesting the existence of alternative biological mechanisms that may be related to prognosis [52].

### 2.7. Differential Diagnosis with Other ALK+ Tumors

Several types of neoplasms other than ALCL also show aberrant ALK expression and/or rearrangements, such as, ALK+ large B-cell lymphoma [11,142,143,144,145]. Other non-hematological malignancies are also ALK positive, such as metastatic non-small-cell lung cancer metastatic carcinoma [146,147], inflammatory myofibroblastic tumors, ALK-positive histiocytosis [148], Merkel cell carcinoma [149], ALK-positive large cell neuroendocrine carcinoma [150], primary/metastatic cutaneous melanomas [151], Spitz tumors with ALK fusions [152], ALK rearranged renal cell carcinoma [153], epithelioid inflammatory myofibroblastic sarcoma [154] and almost all cases of pediatric spindle cell/sclerosing rhabdomyosarcoma [155,156]. In addition, breast, colon, serous ovarian, squamous esophageal, and anaplastic thyroid carcinoma may be positively expressed [156,157].

## 3. Systemic ALK−Negative Anaplastic Large Cell Lymphoma

### 3.1. General and Clinical Aspects

ALK- ALCL is the fourth most common T-Cell Lymphoma, representing 50% of sALCL cases. This entity is defined as a mature T-Cell Lymphoma with uniform, strong expression of CD30, without ALK expression or *ALK* rearrangements [11,12,15]. Patients have a mean age of 54 years and the condition is more prevalent in males (male:female ratio of 1.5) [158]. Patients usually show advanced disease at presentation, referring with B-symptoms [75], frequent BM involvement, and nodal and extranodal affectation (usually in a 1:1 ratio). Approximately 50% of patients present with involvement of single extranodal sites, whereas 25% of cases show multiple extranodal involvement (mainly in soft tissue, mediastinum, bone marrow, liver, spleen, gastrointestinal tract and breast) [75,158].

### 3.2. Morphology and Immunophenotype

Morphology cannot reliably distinguish ALK+ and ALK- ALCL due to their similarities, in addition to the absence of ALK expression or *ALK* rearrangements [11,12,15]. IHC and molecular techniques are essential for establishing a definitive diagnosis in these cases. Observation of mitoses and necrosis is also relevant. Involvement may be focal or intrasinusoidal. The most important diagnostic criterion is the complete or partial infiltration of lymph node or extranodal tissue by “hallmark cells”. Transdifferentiation to malignant histiocytosis has been described on rare occasions [159]. Morphological differences between genetic subtypes will be addressed in the following sections. However, certain features suggest that a case might be ALK−, including the presence of plasmablastic cells and a starry-sky pattern [65] (Figure 5).

Feldman et al. [81] reported that a group of ALK- ALCLs showed upregulation of genes related to several pathways, such as cell cycle, DNA repair, epigenetics and metabolism; the epigenetic pathways involved chromatin-modifying enzymes and histone methylation, unlike what occurs in ALK+ ALCLs. *EZH2* was the most strongly overexpressed gene in this ALK- ALCL cluster, as validated by IHC.

Fewer than 20% of cases express TCR (mainly αβ), and occasional cases lack immunophenotypic evidence of T-cell lineage (“null-cell”) [160,161,162]. Some rare cases with TCR-γδ expression have been reported [163]. CD4 may be expressed in 70% of cases, while CD8 is expressed in only 14% of them [164]. They are positive for CD43, CD2, CD3, CD5 and CD7 (the latter four markers with a frequency of 50%), and cytotoxic markers (TIA1, granzyme B, perforin). They are less commonly positive than the ALK+ subgroup for EMA and Clusterin. CK is usually negative, which contrasts with ALK+ ALCL cases. Immune evasion markers, such as PD-L1, TGF-beta and IL-10, are also frequently expressed. Nuclear pSTAT3 is expressed in 50% of cases. Finally, these cases are negative for Epstein–Barr virus (EBV) infection [165,166]. Rare cases with PAX5 expression (similar to CHL cases) have been reported [167,168,169]. Some of them coexpress PAX5 and CD138 without T-cell markers [170].

Feldman et al. proposed an algorithm based on IHC and FISH assays for genetic subtyping ALCLs. A pSTAT3 (<20%) and LEF1 (>75%) IHC combination showed positive and negative predictive values of 100% each for *DUSP22* rearrangement diagnosis [171]. The complete algorithm for classifying ALCLs is based on IHC of four markers: ALK, LEF1, TIA1 (≥20%), and p63 (≥30%), although FISH assays for *DUSP22* and *TP63* are recommended for ALK− cases. The authors affirmed that this approach is also suitable for pcALCL and BIA-ALCL diagnosis.

### 3.3. Cytogenetic Alterations

Sixty-five percent of ALK- ALCLs show structural chromosomic alterations, most of which are gains at 1q and 6p21. Losses at 17p13 (42%) and at 6q21 (35%), which encode the *TP53* and *PRDM1* genes (the latter playing a tumor-suppressor role in ALCL models, in which it probably acts as an antiapoptotic gene, coding for *BLIMP1*), respectively, have also been observed in 52% of cases [85,172,173]. Cases with losses at 17p13 and/or 6q21 tended to have a worse outcome.

Additional gains of chromosome 2 (Trisomy 2) have been described, but it is not yet known whether these are a primary or secondary event relative to the pathogenesis of ALCL [174]. FISH assays in such cases have identified additional copies of the *PAX5* gene locus [167,168,169].

### 3.4. Molecular Subtypes with Prognostic Significance

ALK- ALCL is a genetically heterogenous entity that encompasses diverse molecular subtypes with respect to the presence or absence of specific translocations of distinct prognostic significance: *DUSP22* rearrangements (DUSP22Rs), *TP63* rearrangements (TP63Rs) and Triple-Negative (TN, with all absent).

#### 3.4.1. DUSP22 Rearrangements

The presence of *DUSP22* rearrangements reported in up to 30% of ALK- ALCL cases [163,175], defines a genetic subtype of ALK− sALCL because of its distinct morphological, phenotypic, genomic and epigenetic features [13]. The t(6;7)(p25.3;q32.3) was the first recurrent translocation to be described in ALK- ALCL, causing the fusion of *DUSP22* at 6p25.3 and *FRA7H* at 7q32.3. This fusion leads to a decrease in the expression of DUSP22 and overexpression of miR29A and miR29B1 at 7q [171,176].

These cases have specific morphological features, such as the frequent presence of “doughnut” cells accompanied by hallmark cells, a sheet-like pattern with medium-to-large cells that lend a monomorphic appearance, and lymph node architecture effacement by neoplastic infiltration by intermediate cells, smaller than those observed in TN ALK- ALCLs and ALK+ ALCLs [177,178]. In contrast to other ALCL subgroups, cases with DUSP22R are generally negative for the cytotoxic markers TIA-1 and granzyme B, being expressed in only 10% and 5% of cases, respectively They frequently show CD3 expression and are negative for pSTAT3 IHC. GEP techniques have identified that DUSP22R is correlated with a strong and uniform pattern of LEF1 expression, with high positive (93.8%) and negative (96%) predictive values for the presence of DUSP22R (Figure 6). Ravindran et al. [179] reported strong LEF1 IHC expression in 15 of 16 DUSP22R cases, confirmed by GEP techniques. Although LEF1 is a nuclear mediator of the Wnt/β-catenin pathway, CTNNB1 RNA and protein levels were not overexpressed in these cases, suggesting that LEF1 overexpression may not be a result of this pathway’s activation.

Several GEP studies have tried to clarify the specific molecular features of this ALCL subtype. Díaz de la Pinta et al. [180] proved that not all cases showing DUSP22/IRF4 rearrangement in FISH assays possess t(6,7)(p25.3;q32.3), which involves either *DUSP22/IRF4* with *LINC-PINT* gene fusion. RNA-seq results indicated that not all fusions showed the same GEP, highlighting underexpression of *TCF3* (*TCF7L1/E2A*), *DLL3*, *CD58* and *BCL2* in those cases showing t(6,7)(p25.3;q32.3). Another recent discovery by Fadl et al. [181] demonstrated that not all *DUSP22* rearrangements are equal, distinguishing between those showing two-color allelic patterns in FISH assays (“normal” rearrangements) and those in which one of the colors of the dual-color probe was missed (“equivocal” rearrangements). The latter cases showed higher levels of expression of pSTAT3 and TIA1 with IHC, and a lower level of LEF1 staining than the former. It is recommended that FISH assays be complemented by IHC for the diagnosis of these cases.

Luchtel et al. [175] demonstrated that DUSP22R cases exhibit molecular peculiarities. DUSP22R cases overexpress cancer-testis antigen (*CTA*) genes such as *CTAG1*, *CTAG2*, *MAGEA10*, *MAGEA5* and *SSX4*, with a marked DNA hypomethylation profile. In addition, DUSP22R cases minimally expressed PD-L1 compared with other ALCLs, but had high levels of expression of *CD58* and HLA class II. Finally, DUSP22R lacked JAK-STAT3 signaling, leading to downregulation of *GZRB* and *ILR2A*. Nevertheless, the cellular signaling acknowledge and the biological implications of these rearrangements including *DUSP22* remain uncertain.

#### 3.4.2. TP63 Rearrangements

Approximately 5–8% of ALK- ALCLs show a rearrangement of *TP63* in 3q28, commonly with *TBL1XR1* due to an inv(3)(q26q28) [163,182,183]. This subgroup has the worst prognosis of all ALK- ALCLs [163].

Morphologically, these cases tend to have more large pleomorphic cells compared with DUSP22R cases [183]. They also show a diffuse, sheet-like growth pattern with apoptotic debris, prominent tingible body macrophages and mitotic figures. Strong and uniform nuclear staining for p63 protein is remarkable, but not specific. The aberrant p63 proteins caused by these rearrangements are known to have oncogenic properties and to inhibit the p53 pathway [184,185,186]. Since these cases are rare, it is difficult to determine further specific histopathological features.

The largest study of p63 protein expression and its association with *TP63* abnormalities in ALCL (116 cases), by Wang et al. [182], demonstrated that p63 was positive in 35% of ALK- ALCLs by IHC. With a positive cutoff value of ≥30%, p63 IHC showed 100% sensitivity for TP63R. The higher frequency of p63 protein expression compared with that of the *TP63* rearrangement (35% vs. 8%, respectively) is explained by the presence of extra copies of *TP63* in cases without rearrangement that probably arise through copy number gains. Interestingly, extra copies of *TP63* are associated with extra copies of *DUSP22*, which could be explained by aneuploidy rather than specific focal gains. This work demonstrated that IHC for p63 is not specific to TP63R but is useful for selecting candidates for FISH assays.

#### 3.4.3. Triple-Negative ALK- ALCL

Triple-Negative ALK- ALCL (TN) is defined by the lack of the previously described alterations involving *ALK*, *DUSP22* and *TP63*. This subtype is the most common type of ALK- ALCL, accounting for 40–60% of cases [81,163,187].

These cases can be divided into those that are positive and negative for pSTAT3. TN pSTAT3+ cases present sheet-like neoplastic cells, large pleomorphic cells scattered in a lymphocyte-rich background, cytotoxic molecules, epithelial membrane antigens and PD-L1, but are negative for CD3 and CD5. Conversely, TN pSTAT3- cases show pleomorphic neoplastic cells with monomorphic inflammatory background and generally lack cytotoxic phenotype markers [188].

#### 3.4.4. New Genetic Approaches and Novel Potential Subgroups in ALK- ALCL

The current classification for ALK- ALCL may change due to recent findings that may define novel genetic subgroups (Table 4). These include the following:

##### “Double Hit” Cases with DUSP22 and TP63 Rearrangements

Five rare cases described by Karube [189] and Klairmont [190], known as DH, harbored rearrangements in *DUSP22* and *TP63* genes. These cases showed diffuse, cohesive infiltrate by large atypical lymphoid cells, occasionally with kidney or horseshoe-like nuclei. These tumor cells were positive for CD2, CD3, CD30 and p63, and negative for CD7, CD8, CD15, CD20, ALK and cytotoxic markers (TIA-1, granzyme B and perforin). The survival outcomes of this rare clinicopathological entity are not known and are a matter of controversy in the literature, needing more evidence and case reports to enable them to be clarified.

##### JAK2 Rearrangements and Morphology Variants

Fitzpatrick et al. [191] described six cases of ALK- ALCL with *JAK2* rearrangements within a cohort of 97 samples (6%) detected by Next Generation Sequencing (NGS). The most remarkable case carried a novel *PABPC1::JAK2* fusion and was associated with unusual CHL-like features. The other five cases showed *JAK2* rearrangements with four different novel partners (*TFG*, *PABPC1*, *ILF3* and *MAP7*) and with one previously described gene (*PCM1*). With respect to their morphology, all of them showed CHL characteristics and 80% were unusually positively stained for CD15 (Figure 7).

##### ERBB4 Expression Subclass Morphology

In the study by Scarfó et al. [192], GEP data were collected from 249 cases of TCLs and normal T-cells. Ectopic coexpression of *ERBB4* and *COL29A1* genes was found in 24% of ALK- ALCL patients and was subsequently confirmed by Western blot and IHC. This could prove the existence of a new subclass of ALK- ALCL characterized by aberrant expression of ERBB4-truncated transcripts carrying 59 intronic untranslated regions.

##### FRK Fusions

Novel fusions with the *FRK* gene (6q22.1) have been described in ALK- ALCL, with a frequency of 5.4% in the studied cohort [193]. The known fusion partners are *PABPC1* (8q22.3), *MAPK9* (5q35.3) and *CAPRIN1* (11p13). The *CAPRIN1::FRK* fusion transcript was functional, with a high level of expression of the chimeric protein that contributed to the rise in pSTAT3 levels in vitro. It was possible to target these cells using the kinase inhibitor dasatinib, thereby demonstrating that FRK rearrangements represent a potential therapeutic target.

##### MYC Rearrangements May Be Associated with Poor Prognosis

Two patients harboring *MYC* rearrangements, observed with FISH assays, experienced a worse clinical course, in which they progressed rapidly during aggressive treatment. Although both cases presented advanced-stage diseases and were elderly, the association between this novel *MYC* rearrangement and poor prognosis is interesting, particularly since one of the patients showed DUSP22R, which generally has better outcomes [194].

### 3.5. Signaling Alterations, Pathogenesis Mechanism and Prognostic Significance of STAT3 Activation

Even though ALK- ALCL pathogenesis is unrelated to ALK activation, ALK+ and ALK- ALCLs share a STAT3-mediated oncogenic mechanism. Therefore, it is possible that JAK/STAT3 pathway inhibitors have a therapeutic application not only in ALK+ ALCL, but also in ALK- ALCL [108].

Two mutually exclusive mechanisms lead to the constitutive activation of STAT3 in ALK- ALCL [195]:Oncogenic point mutations (~20% of cases) in the JAK1 kinase domain (G1097D/S, L910P) [195] and/or the STAT3 SH2 domain (Y640F [196], N647I, D661Y [197] and A662V [195]).Oncogenic fusion genes displaying concomitant transcriptional and kinase activities capable of sustaining the ALCL phenotype via STAT3 activation, such as *NFκB2::ROS1*, *NCOR2::ROS1*, *NFκB2::TYK2*, and *PABPC4::TYK2* fusions [195].

Nevertheless, pathogenesis and the molecular mechanisms involved in ALK− are not clearly understood, further research being needed to clarify them. However, the mechanisms that lead to the activation of each ALCL subtype are different, and a recent study demonstrated that JAK/STAT3 signaling activation alone was not sufficient to promote cell survival, and that the activation of cytokine receptor signaling was required in ALK- ALCLs [198] (Figure 8).

The association between the activation of this pathway and clinical outcomes is not yet fully understood, but several studies have identified potential mechanisms that could explain this.

### 3.6. Role of Non-Coding RNAs in ALK- ALCL

The role of non-coding RNAs in ALK- ALCL and their potential pathological function is largely unknown, with only a few studies having been undertaken.

One of the most important transcripts of this nature is BlackMamba, a novel lncRNA associated with ALK- ALCL, which may help maintain the neoplastic phenotype in these cells [199]. Another lncRNA implicated in complex transcriptional regulation is MTAAT, favoring the progression of ALK- ALCL [200].

Finally, MIR503HG (miR-503 host gene) is highly overexpressed in ALCL ALK− cell lines. In vitro and in vivo assays proved that this miRNA enhances tumor cell growth through the miR-503/Smurf2/TGFBR axis [201].

### 3.7. Mutational Landscape of ALK- ALCL

Luchtel et al. [202] described a novel mutation in the Musculin (*MSC*) gene, almost exclusively in DUSP22R cases (except for one TN), that could specifically explain the activation of MYC and cell cycle progression on them. This alteration promotes the activation of CD30–IRF4–MYC axis. Wild-type MSC acts by repressing MYC and cell cycle progression, thereby activating *E2F2* transcription and regulating lymphocyte development. This mutation (MSC^E116K^) prevents the binding of this basic helix–loop–helix transcription factor to its DNA-specific sequence, impeding its function, and favoring positive feedback with itself and IRF4 protein.

*TP53* is mutated in a small proportion (approximately 16%) of ALCL tumors. Approximately 23% of ALK- ALCLs carry mutations in this gene, which is usually associated with a poor prognosis [88,89]. Nevertheless, the level of p53 expression is usually high in them, and may be functional in these tumors [203] (Figure 9).

Another relevant mutation is that occurring at the *PRF1* gene (which encodes for perforin), highlighting the germinal PRF1^A91V^ variant, especially in pediatric ALCL patients [204]. Finally, some other mutations with no clear clinical significance are located in the *PRDM1* (13.5%), *EPHA5* (16%), *LRP1B* (11%), *KMT2D* (11%) [88], *BANK1*, *FAS* and *STIM2* genes [195].

### 3.8. Prognosis and Prediction

ALK- ALCLs show worse outcomes than ALK+ ALCLs, with 5-year OS rates of 32% [27,54], but prognosis differs markedly among the subgroups comprising this entity.

Positivity for all cytotoxic markers (TIA-1, granzyme B and perforin) is an unfavorable prognostic indicator [163,205].

Liang et al. [45] identified an association between strong IL2R expression in ALCL patients and aggressive clinical presentation. In vitro and in vivo assays demonstrated the importance of the BATF3/IL2R axis module for ALCL biology and identified IL2R targeting as a promising treatment strategy for ALCL.

Lobello et al. [88] presented data proving that *STAT3* and *TP53* mutations are associated with poor prognosis in ALCL. *TP53* was the most recurrently mutated gene (23% ALK- ALCL cases). Alterations in *STAT3* and *JAK1* were only present in ALK- ALCL, but only the former was found to be the best predictor of OS in this subgroup.

Two separate studies initially established that the presence of *DUSP22* rearrangements was associated with a favorable prognosis, similar to ALK+ ALCL cases, even in the absence of BM transplant treatment consolidation [163,206]. For example, in the NLG-T-01 study [207], DUSP22R patients associated tended to have a very good outcome and a 5-year OS of 83% after undergoing HDT/ASCT. These are similar results to those obtained from the Mayo and Danish cohorts [208] (90% and 80%, respectively, without undergoing this therapeutic scheme). This prompted the inclusion of a note in the 2018 NCCN Guidelines [209] that mentioned the possibility of adopting ALK+-like regimens in DUSP22R cases. Nevertheless, subsequent studies produced contradictory results, identifying cases with a poorer prognosis than expected. Hapgood et al. [187] identified high-risk DUSP22R patients. The outcome of these cases was worse than observed in previously published series (5-year OS of 40% and 5-year Progression-Free Survival [PFS] of 45%). In fact, equivocal DUSP22R patients also exhibited different clinical behavior, with outcomes intermediate between those of Normal DUSP22R cases and other ALK- ALCLs without *DUSP22* rearrangements [181]. More recently, Qiu et al. [210] reported did not have different clinical outcomes from those of other ALK- ALCL subgroups. From these findings, we conclude that further studies are needed to clarify the controversy around DUSP22R clinical behavior.

Vasmatzis et al. described the clinical features of TP63R cases in a cohort of 190 PTCLs [183]. TP63R was associated with poorer OS on average than among the overall cohort of PTCLs (17.9 vs. 33.4 months, respectively).

TN cases have clinical outcomes intermediate between those of TP63R and DUSP22R cases [175,208]. A study by Wang et al. [188] established that TN cases with pSTAT3 expression had better outcomes than pSTAT3– patients (5-year OS of 50% vs. 20%, respectively).

### 3.9. Differential Diagnosis: ALK- ALCL vs. CD30+ PTCL-NOS

A huge effort has been made in recent years to identify a gene expression profile specific to ALK- ALCL that would allow cases initially diagnosed as PTCL-NOS to be reclassified, as well as enabling the search for differences from other TCL [54,191].

The distinction between ALK- ALCL and PTCL-NOS with large cells and CD30 expression (CD30+ PTCL-NOS, approximately 10% of cases) is not only challenging but is also prone to subjectivity. Typically, a differential diagnosis between the two tumor types is made when the morphology and immunoprofile of the sample do not match the features previously described for ALCL ALK- (Table 5).

Rearrangements of *DUSP22* or *TP63* genes favor a diagnosis of ALCL ALK- over CD30+ PTCL-NOS, but a very small subset of PTCL-NOS may also harbor them, ruling them out as features relevant to a differential diagnosis. For example, Pedersen et al. [208] presented a rare case of CD30+ PTCL-NOS with some ALCL-like features that harbored rearrangements of both *DUSP22* and *TP63.*

The last decade has seen the development of predictive models and gene markers used to discern pathogenic differences between ALK- ALCL and PTCL-NOS. However, the techniques used are costly and unsuitable for routine practice. In one study, by Iqbal et al. [211], GEP was performed on 372 PTCL samples, leading to the identification of robust molecular classifiers based on the biology of tumor cells and their microenvironment. Thirty-seven percent of the cases diagnosed as PTCL-NOS were reclassified into other subtypes on the basis of their molecular signatures. This demonstrates the urgent need for further research to diagnose these patients accurately in the clinical milieu based on their molecular features.

A study by the European T-Cell Lymphoma research group [212] validated a three-gene model (*TNFRSF8*, *BATF3*, *TMOD1*) that was mostly expressed in ALK- ALCL, as revealed by RT-qPCR. This signature was able to distinguish the two tumor subtypes with an accuracy of ~97%. Other overexpressed genes in ALK- ALCL were *CD80*, *DC86*, *CCND2* and miR155HG. In addition, the *CCR7*, *CNTFR*, *IL22* and *IL21* genes had previously been described as being overexpressed [213].

Chromosomal imbalances observed in ALK- ALCL also differ from those identified in PTCL-NOS, especially losses of 5q (26%) and 9p (31%) in PTCL-NOS but not in ALK- ALCL [172].

Finally, a recent work by Xiang et al. [214] demonstrated that pSTAT3-Y705/S727 IHC can be used to distinguish between ALK- ALCL from CD30+ PTCL-NOS. pSTAT3-Y705 is mainly expressed by lymphoma cells, while pSTAT3-S727 positivity comes from background tumor-infiltrating lymphocytes. The former had greater specificity, while the latter proved to be more sensitive in the differential diagnosis of the two entities. Nevertheless, the authors recommend the use of pSTAT3-S727 IHC due to the low proportion of ALK- ALCL (13%) expressing pSTAT3-Y705 at a very low level (<30%). A high level of expression of pSTAT3-S727, with a sensitivity of 0.86 and specificity of 0.9, appears to be a promising biomarker for the differential diagnosis of CD30+ PTCL-NOS and ALK- ALCL.

## 4. Breast Implant-Associated Anaplastic Large Cell Lymphoma

### 4.1. General and Clinical Aspects

Breast Implant-Associated Anaplastic Large Cell Lymphoma (BIA-ALCL) is a mature CD30-positive T-Cell Lymphoma that arises in relation to a breast implant, or, in extremely rare cases, may be caused by a gluteal implant [215]. ALK- ALCL profile is the most common presentation of this pathology. BIA-ALCL has recently been recognized as a distinct entity, defined as a site-specific lymphoma and the most common lymphoma type associated with breast implants [12,15]. In 2016, BIA-ALCL was included as a provisional entity in the WHO classification [11], finally being included as a definitive entity in the provisional 5th revision of the WHO classification [12].

The geographical incidence varies, with more reported cases in the United States, Europe and Australia than from the rest of the world, as do the risk estimates (from 1/3000 to 1/30,000 women with implants globally) [216]. The use of textured breast implants, compared with smooth-surfaced prostheses, brings a higher risk of developing this disease, whereby approximately 75% of cases present with implants of this type of material [217].

Patients tend to debut during the fifth decade of life [218,219,220]. Most cases debut 10 years after implant placement (with a mean age at implantation of 42 years) [217], typically presenting with a unilateral effusion of up to 700 mL [219,221]. Bilateral affectation has been observed in 5% of cases [222]. The incidence between 14 and 16 years after surgery is higher (3.31/1000 cases) [223]. Incidence in cases associated with cosmetic indications and with large breast populations is similar [216].

BIA-ALCL is defined by a layer of tumor cells caught in a fibrinoid and extensive necrotic meshwork along the luminal side of the capsule [12,224]. It is usually restricted to the peri-implant space or as superficial deposits on the luminal side of the peri-implant fibrous capsule at presentation. However, in 10–30% of cases, it presents as a tangible mass within or beyond the capsule, infiltrating the surrounding soft tissue, skin or breast parenchyma [225]. In addition. necrosis is usually prominent at early and advanced stages [221,226]. Patients with BIA-ALCL experience a painful effusion around the implant. Regional LNI is present in 20% of patients, and lymphadenopathy may be the first manifestation of the disease (representing the main reason for misdiagnosing this disease as CHL or sALCL) [227]. When present, from a histological point of view, interfollicular or perifollicular involvement that is usually sinusoidal, but less frequently diffuse, is observed [228]. B symptoms, capsular contractures and cutaneous lesions have been described on rare occasions, and some patients have even been asymptomatic [224].

Disseminated disease at presentation is rare; the few reported cases underwent aggressive treatment. Silicone material can be identified focally throughout the capsule, usually with giant cell reaction [229,230,231].

### 4.2. Morphology and Immunophenotype

Cytological aspirates of seroma fluid surrounding the affected implant are the main tool for the primary diagnosis of BIA-ALCL. Cytocentrifugation and filtration of this effusion fluid, followed by the preparation of a cell block, are recommended. Fixation of the capsulectomy specimen and selection of multiple representative sections to assess capsular invasion and tumor staging should be performed. Excisional lymph nodes biopsies are the most effective means of assessing LNI [232].

Tumor cells show abundant basophilic and vacuolated cytoplasm. Nuclei are large and pleomorphic, oval or lobated. Atypical mitotic figures can be seen [226,233]. Malignant cells are large, with pleomorphic and anaplastic morphology and tend to form cohesive clusters. Affectation of regional lymph nodes is not frequent, and, when present, involvement patterns are typically sinusoidal, perifollicular, interfollicular or even diffuse [12,234].

The immunophenotype is similar to that of ALK- ALCL, featuring uniform, strong expression of CD30, and negative ALK expression. The markers CD3, CD5, CD8 and CD7 are usually negative [235]. On the other hand, neoplastic cells are positive for CD4, CD43, CD25, MUM1 and GATA3, as well as for cytotoxic markers (TIA1, granzyme B, and/or perforin). Loss of the TCR antigen is variable, and, if expressed, could be alpha/beta or gamma/delta [236]. *TCR* genes are commonly clonally rearranged (80% of cases) [219]. CD8+ and double-negative CD4/CD8 cases are rare, but some have been described [224,226]. Nuclear pSTAT3 expression, consistent with the molecular biology of the tumor, is also remarkable. EBV-encoded small RNA (EBER) and LMP1 are negative. EMA staining is variable, and its positivity is less common than in ALK- ALCL [15,65,219,237,238] (Figure 10). Finally, the detection of cytokines in seromas could be useful: IL-9, IL-10, IL-13, IL-22 and IFN-γ (but not IL-6) distinguish malignant from benign effusions [239].

### 4.3. Pathogenesis of BIA-ALCL

Two main theories have been established to account for BIA-ALCL pathogenesis: a bacterial infection (which leads to chronic inflammation), and a hypoxia-related origin (Figure 11).

Some authors have postulated that BIA-ALCL begins with the stimulation driven by silicone and a periprosthetic bacterial biofilm, which causes a Th17 cell response induced by IL-10, IL-13 and bacterial lipopolysaccharide antigens. This causes deregulation of T-cells, favoring cellular stress and oncogenic events [240,241,242,243]. Kadin et al. [244] analyzed cell lines and clinical samples of BIA-ALCL using GEP techniques, flow cytometry, ELISA and IHC to characterize transcription factor and cytokine profiles. This study demonstrated that BIA-ALCL shares many features with pcALCL, such as the common expression of *SOCS3*, *JunB*, *SATB1*, and a cytokine profile suggestive of a Th1 phenotype. These findings support the idea that BIA-ALCL arises from chronic bacterial antigen stimulation of T-cells.

The other hypothesis is based on the metabolic changes observed in BIA-ALCL samples, which are related to the activation of hypoxia-related factors. Oishi et al. [245] presented RNA-seq data that showed upregulation of hypoxia signaling genes, highlighting the hypoxia-associated biomarker carbonic anyhydrase-9 (CA9). The results were validated by IHC in all samples. Furthermore, in vitro studies revealed that CA9 was induced after forcing cells to grow under hypoxic conditions. The authors also presented results from silencing and overexpressing assays in mice that supported the hypothesis that BIA-ALCL is a hypoxia-associated neoplasm. In cases with extramammary presentation, this marker helps to establish a differential diagnosis from a true systemic ALK- ALCL, which does not express it.

### 4.4. Molecular Alterations

Several mutations and genomic alterations that predispose to BIA-ALCL have been described, such as germline mutations in *TP53* and/or in *BRCA1/2*. These occur in 1/1200 patients [246,247].

As in the cases of ALK+ and ALK- sALCL, constitutive activation of the JAK/STAT3 pathway is one of the major pathogenic events. It could arise from mutations occurring in *STAT3*, *STAT5B*, *JAK1* and *JAK2*. Loss-of-function mutations in *SOCS1* and *SOCS3*, and/or genomic amplifications of JAK/STAT pathway genes have also been reported. Point mutations in epigenetic modifiers known to be critical in this pathology (i.e., those involving *KMT2C*, *KMT2D*, *CHD2*, *CREBBP* and *DNMT3A*) are also a risk factor [246,247,248,249,250]. In addition, the *STAT3::JAK2* fusion could support the activation of the JAK/STAT pathway [234,236,251]. *EOMES* and PI3K-AKT/mTOR mutations have been identified, along with other epigenetic regulator genes, such as *TET2* [247]. Some reported cases showed focal amplification of *PD-L1* (*CD274*) at 9p24.1, acting in synergy with constitutive pSTAT3 signaling, which favors the tumor’s immune escape [252].

Finally, Los-de Vries et al. observed that the loss of 20q13.12–13.2 is highly specific to BIA-ALCL [253]. Another relevant finding of this work is that there are differences between BIA-ALCL seroma and BIA-ALCL tumor regarding Copy Number Alterations (CNAs). BIA-ALCL seroma showed a significantly higher CNAs and heterogeneity of them. These data suggest that there is a greater diversity of subclones in the seroma, which undergo clonal upon infiltration in the breast parenchyma. Table 6 summarizes the main molecular alterations of BIA-ALCL.

Regarding gene expression peculiarities, in vitro assays in TLBR1 and TLBR2 cell lines demonstrated that STAT6 may be relevant in BIA-ALCL signalling, whereby it acts via IL-4Rα and IL-13Rα1 receptors (and the release of IL-4 and IL-13, respectively) [254].

### 4.5. BIA-ALCL: Staging, Subtypes, Treatment Approaches and Prognosis

BIA-ALCL cases are rare, and most of the studies that have analyzed their clinical aspects and tried to establish prognostic subgroups have been retrospective [219,255]. Nevertheless, their authors agree that the crucial factor is the presence or absence of infiltration beyond the capsule. Various staging guidelines have been proposed, highlighting the classic classification based on the Lugano revision of the Ann Arbor system [256,257]. However, the alternative TNM (tumor, lymph node, metastasis) disease-specific modded staging system, which allows BIA-ALCL cases to be classified into a broader range of options, has been validated and included in the NCCN guidelines for BIA-ALCL [258,259]. This system was developed by researchers at MD Anderson Cancer Center [260,261]. Stage is determined by a PET/CT scan, which must be performed pre-operatively in any confirmed case of BIA-ALCL [258,262]. Using this modified staging system, most patients with BIA-ALCL have low-stage disease, with 83–96% of patients classified as Stage I variants [229,259,261]. The staging in the TNM system is as follows:IA: malignant cells are confined to the fluid or form a layer on the luminal side of the capsule.IB: early capsule infiltration is observed, but cells are confined to the internal capsule.IC: cell aggregates or sheets may infiltrate the capsule.IIA: cell infiltrates beyond the capsule.IIB: involvement of one regional lymph node.III: multiple regional lymph nodes involvement.IV: spread to other organs and distant sites.

BIA-ALCL cases can be classified into two major subgroups depending on the diagnosed stage at presentation:**In situ BIA-ALCL (Stage I):** anaplastic cell proliferation is confined to the fibrous capsule. Patients have an indolent clinical course and generally remain free of disease after capsulectomy and implant removal.**Infiltrative BIA-ALCL (Stage II and beyond):** pleomorphic cells massively infiltrate the adjacent tissue. These cases have a more aggressive clinical course that may require aggressive therapy in addition to implant removal (justifying cytotoxic chemotherapy).

Most patients have an excellent outcome. Surgery is the cornerstone of therapy, with complete excision of the capsule and implant, leading to complete remission and 5-year OS rates of almost 100% in patients with in situ BIA-ALCL. The 5-year OS rate is lower (approximately 50–70%) in infiltrative BIA-ALCL patients, who normally show advanced disease with a non-resectable mass with LNI or disseminated disease at diagnosis. In these latter cases, approved therapy for systemic nodal T-Cell Lymphomas is indicated [255].

Support therapy consists of CHOP or CHOP-like chemotherapy regimens. However, the study by Miranda et al. [255] which reviewed the literature for all cases of BIA-ALCL published between 1997 and 2012, and updated their clinical follow-up, found the OS or PFS to be similar between patients who did and did not receive chemotherapy. This casts doubt on there being a clinical advantage to using this kind of treatment, instead making watchful waiting an option until further treatment is needed. Capsulectomy and implant removal were performed on 93% of the patients and 78% received systemic chemotherapy. Of the 16 patients who did not receive chemotherapy, 12 opted for watchful waiting and four received only radiation therapy.

Laurent et al. [219] collected 19 cases between 2010 and 2016. Their immunomorphological features, molecular data and clinical outcome were retrospectively analyzed. Implant removal was performed in 17 of the 19 patients, with additional treatment based mostly on CHOP or CHOP-like chemotherapy regimens (10 out of 19 patients) or irradiation (one patient). Patients with in situ BIA-ALCL had a 2-year OS of 100%, in contrast to the 52.5% for infiltrative BIA-ALCL.

Ferrufino-Schmidt et al. [228] observed that in a cohort of 70 patients (cases reported from 1997 to 2016 in the Department of Hematopathology, University of Texas MD Anderson Cancer Center), 20% of them had LNI at diagnosis, mostly with an axillary location and a sinusoidal pattern (except two cases with Hodgkin’s-like patterns). These cases had worse clinical outcomes, serving to indicate that infiltrative BIA-ALCL is associated with a higher risk of LNI, similar to what Miranda et al. [255] concluded. In the latter study, 93% of patients with in situ BIA-ALCL confined by the fibrous capsule achieved complete remission, compared with 72% of the cases with infiltrative BIA-ALCL, which had worse OS and PFS.

### 4.6. Differential Diagnosis with Other Entities

In certain cases, BIA-ALCL pleomorphic neoplastic cells show Reed–Sternberg-like features and are embedded in a rich eosinophilic microenvironment. Clinicopathological data should be examined simultaneously to avoid a misdiagnosis of Hodgkin’s lymphoma [228].

Although most Breast Implant-Associated Lymphomas are BIA-ALCLs, a new category of rare B-cell lymphomas related to these prostheses, known as fibrin-associated large B-cell lymphomas (FA-LBCLs), has emerged. These cases are EBV+ DLBCL, and also express CD30. Given their positivity for EBV and breast implant association, these cases could be specifically renamed as BIA-FA-LBCLs. Their differential diagnosis with BIA-ALCL is challenging due to the clinical similarities of the two entities [263].

Turning now to BIA-FA-LBCL histology, morphology is very similar to BIA-ALCL, with large, atypical cells expressing CD30. Nevertheless, B-cell markers are expressed, such as CD20 (positive in most cases), PAX-5 and CD79A, along with EBER. T-cell markers, PD-L1 expression and *MYC* rearrangements have been described on rare occasions [264,265].

Finally, one differential diagnosis that is challenging to establish is that involving BIA-ALCL and Breast Implant-Associated Squamous Cell Carcinoma (BIA-SCC). The clinical presentation of the two entities is very similar, but BIA-SCC is generally more aggressive, and patients have worse clinical outcomes (6-month mortality rate of 43.8%) [266,267,268]. Various techniques, such as NGS, can help distinguish between BIA-ALCL (with known mutations) and BIA-SSC. Flow cytometry is very useful, being able to detect T-cells CD30+ ALK− in BIA-ALCL and squamous cells (CK 5/6+ and p63+) in BIA-SSC.

## 5. Updated Perspectives in Systemic ALCL

One of the most important studies of ALCLs has recently been carried out by Feldman et al., in which 689 cases underwent expert consensus review as part of the Lymphoma/Leukemia Molecular Profiling Project (LLMPP) [81]. Genetic subtyping assays (IHC, FISH) for ALK, DUSP22, TP63 and TN, in addition to pSTAT3 IHC, were performed on all samples. RNA-seq was performed and evaluable in 393 cases. OS data were available for 257 sALCL patients, these indicating a favorable prognosis associated with DUSP22R (5-year OS of 95%) and ALK+ ALCL (87%), an intermediate prognosis for TN cases, and a poor prognosis for TP63R and DH patients (0%). The authors identified two main molecular types of ALCLs using unsupervised GEP techniques: Types I and II. Type I was associated with pSTAT3 expression and featured enrichment of genes of the JAK-STAT3 pathway and its related pathways, such as TNFα-NFκB signaling. Type I ALCLs included ALK+ ALCLs, BIA-ALCLs and a subset of TN ALCLs. Type II included DUSP22R, TP63R, DH and the other TN cases. This cluster was enriched for cell-cycle, DNA repair, epigenetic and metabolic pathway genes, highlighting the importance of the activation of epigenetic pathways (such as chromatin-modifying enzymes and histone methylation). This work is relevant because, in a large cohort without substantial missing clinical data, it shows that pSTAT3 IHC is >90% accurate as a surrogate for GEP-based subtyping and can be applied using FISH and IHC in routine practice. This simplified molecular classification is suitable for diagnosis, prognosis, and could help select the best therapeutic option for each patient.

## 6. Primary Cutaneous Anaplastic Large Cell Lymphoma (pcALCL)

### 6.1. General Aspects of pcALCL

Primary Cutaneous Lymphomas with CD30 expression constitute a spectrum of lesions, from pcALCL to Lymphomatoid Papulosis (LyP). Although they share a common hallmark, positive CD30 expression, they differ in their clinical presentation and histological characteristics [269]. LyP and pcALCL are both included in the primary cutaneous T-cell lymphoid proliferations and lymphomas disorders of 2022’s WHO-HAEM5 [12]. LyP is most frequently found in trunk and extremities [55] with mucosal involvement or concurrent mucosal and cutaneous sites [270], while pcALCL is a primary cutaneous disease that can be found in the skin at any location [271], with single or multiple lesions arising at the same site. It is considered the second most common Cutaneous T-Cell Lymphoma [56].

In terms of their clinical features, LyP exhibits multiple self-resolving lesions that occur in outbreaks in different stages of development. For instance, the type E variant usually forms ulcerated lesions with crust in acral areas. The size of the lesions may vary and does not define the entity. In contrast, pcALCL is characterized by a single stable lesion or multiple lesions at a unique location, but at the same stage. It usually does not ulcerate, although this can occur in some cases.

### 6.2. LyP Morphology and Immunophenotype

The histological spectrum of LyP is especially broad and several subtypes are recognized: A, B, C, D, E and lymphomatoid papulosis with DUSP22 *locus* rearrangement. Other rare subtypes have been described, including folliculotropic, syringotropic and granulomatous [272]. Subtypes A, B and C mostly express a CD4+ phenotype, while subtypes D and E express the CD8+ immunophenotype [273].

The DUSP22R subtype shows a particular biphasic phenotype, with either CD8+ or double-negative CD4/CD8. The use of LEF1 in combination with cytotoxic markers, especially TIA1, can serve as surrogate biomarkers to identify this specific subtype. A positive LEF1 phenotype in the absence of TIA1 or STAT3 may indicate the existence of the *DUSP22* rearrangement [171]. Rare cases have been reported for a CD56+ phenotype, expression of TCR gamma/delta [274,275] or a follicular T-helper phenotype [276].

### 6.3. pcALCL Morphology and Immunophenotype

A brief histopathological description of this pathological entity includes an anaplastic cell morphology with round, oval and irregularly shaped nuclei featuring prominent nucleoli [56]. From an immunohistochemical point of view, cells exhibit a CD4+ phenotype, a variable loss of CD2, CD3, CD5 and CD7 markers, but can also present CD4-/CD8+ or CD4+/CD8+ immunoreactivity. CD30+ is expressed in more than 75% of the cases, whereas EMA is less frequently positive in pcALCL [277]. An increased expression of NOTCH1 has also been described in pcALCL [278] (Table 7).

Other relevant variants of pcALCL include neutrophilic/eosinophilic rich [279], angiocentric or angiodestructive [280], epidermotropic or *DUSP22*-rearranged variants. The latter possesses large cells in the dermis and small intraepidermic cells that tend to exhibit the CD8 or double-CD4/CD8-negative immunophenotype and is able to invade lymph vessels [177]. Other less common variants include small-cell morphology, apoptotic bodies, lymphohistiocytic morphology, lymphovascular invasion, intravascular localization, keratoacanthomatous hyperplasia, myxoid stroma, and sarcomatoid morphology [281].

### 6.4. Intralymphatic CD30+ Large T-Cell Lymphoma Morphology and Phenotype

Another entity that may involve the skin is intravascular cutaneous ALCL, with anaplastic tumor cells expressing CD4 and CD30, while are negative for EBER. It also needs to be differentiated from the intralymphatic spread of CD30+ tumor cells in the surrounding tissue [282]. In fact, ALK- ALCL and related CD30+ ALK− T-cell lymphoproliferative disorders involving the lymphatics are part of the expanding spectrum of CD30+ T-cell lymphoproliferative disorders Lymphatic vessels express CD31 and are negative for CD34, in contrast to intravascular T-Cell Lymphoma, which affects blood vessels and may also show CD30 positivity [283].

Benign atypical intralymphatic CD30+ T-cell proliferation (IPTCLB) is another rare condition that mimics intravascular lymphoma, usually exhibiting a CD4- and CD30+ immunophenotype with large, blastoid T lymphocytes within lymphatic vessels, and that is associated with common inflammatory conditions [284].

Intravascular T-Cell Lymphoma is characterized by a cytotoxic phenotype and its location within blood vessels. It may or may not express CD30 and positivity for EBV [285].

### 6.5. Molecular Alterations

It remains crucial to review the most recent molecular findings that have been identified in up-to-date studies using whole-genomic profiling to obtain a more accurate mutational signature of pcALCL in order to gain insight into its pathogenesis.

#### 6.5.1. Chromosomal Rearrangements

*DUSP22* rearrangement remains the most common genetic abnormality in pcALCL, located on chromosome 6p25.3 and observed in 20–25% of pcALCL cases [286]. The gene most commonly associated with the *IRF4* or the *DUSP22* fusion is *LINC-PINT* [180]. A decrease in *DUSP22* protein expression, which reduces STAT3 activation, as well as an inactivation of the LCK pathway, can occur [281]. On the other hand, the morphology of pcALCL with the *DUSP22* gene rearrangement differs from that of sALCL that infiltrates the skin and exhibits the same translocation [180].

Clonal rearrangements of the *TCR* can also be identified and be present in the majority of cases in children and adults [287,288]. Clonality of *TCR* is found in a minority of LyP specimens, unlike in pcALCL cases [289].

A novel chimeric fusion, *NPM1::TYK2*, was also described in 15% of cases [290]. Other gene fusions have been described, although they are considered rare. Although a small number of cases express ALK+, it can be plausible to identify in such cases a *NPM1::ALK* gene fusion, as well as additional gene partners of the *ALK* gene, such as *TRAF1*, *ATIC* and *TPM3* [291].

Other gene fusions include the *TP63* gene, located on 3q28, which is associated with a poor survival in ALK− sALCL, but is rare in pcALCL [292]. Recent Whole-Genome Sequencing (WGS) studies have reported four genes that are recurrently rearranged in pcALCL, in addition to *DUSP22* and *TP63* [293].

#### 6.5.2. Copy Number Alterations

Secondary genetic alterations, including CNV, have been observed in pcALCL. In fact, numerous chromosomal gains at 7q31 and 17q, as well as losses at 3p, 6q and 13q have been reported in pcALCL [281]. Comparative genomic hybridization (CGH) studies of this pathology have demonstrated a type of chromosomal instability in various genes involved in the MAPK pathway, such as *NRAS* (1p13.2) and *RAF1* (3p25), as well as *MYCN* (2p24.1) and *FGFR1* (8p11) [294]. Recent studies using WGS technology in a series of five cases of pcALCL detailed the loss of *MAP2K3*, *STAT2–3* and *DNMT3* genes [295]. Other WGS studies in a larger series of pcALCL cases (n = 12) exhibited losses at 3q, 6q, 7p/q, 8p, 13q, 16q and even loss of the Y chromosome, while gains were observed at 1q, 2p/q, 7q and 12q. It is of note that deletions at 6q21 were the most frequent CNA. This region contains the *PRDM1* gene, which encodes a transcription factor that attenuates T-cell proliferation and survival. Importantly, gains in the 1q region, which contains the *TNFRSF8* gene, encode the CD30 receptor, overexpression of which is a hallmark of pcALCL. Other recurrent CNAs affecting the *EZH2*, *RBFOX1* and *STK24* genes were validated by droplet PCR [47,296]. Loss of 9p21, which contains the *CDKN2A* locus, encoding the p14, p15 and p16 proteins involved in cell cycle control, has rarely been described in pcALCL [297].

#### 6.5.3. Small-Nucleotide Variants

Recurrent point mutations can also be associated with this entity. Whereas JAK/STAT mutations leading to STAT3 overactivation are seen in sALCL, no small-nucleotide variants (SNVs) were found in pcALCL, except for two cases of cutaneous ALCL with the p.Y640F and p.G656C mutations (Table 8). Conversely, WGS studies have identified several genes presenting indel mutations, such as *CDK14*, *CNOT1*, *CREBBP*, *EOMES*, *KMT2A*, *KMT2D*, *LRP2*, *NOTCH1*, *PDPK1*, *SETD2* and *SMARCA4*, as well as tumor-suppressor genes like *CSMD1*, *LRP1B* and *PIK3R1* [47,295].

A recent study demonstrated a plausible association between the *DUSP22* rearrangement and MSC^E116K^ mutation in a cohort of pcALCL that also presented a LEF1+/TIA1- immunophenotype in almost every case [298] (Figure 12).

#### 6.5.4. Role of Non-Coding miRNA

Additional differentially expressed miRNAs have been identified in pcALCL, such as miR21, miR27b, miR29b, miR30c, miR142 and miR155 (upregulated), and miR141 and miR200c (downregulated) [299]. In fact, miR29 targets *BCL2* promoter, causing *BCL2* downregulation in myeloid leukemia and lymphoma [300]. Actually, miR155 plays a key role in T-cell response by regulating CTLA4. Another onco-miRNA, such as miR142, may act as a suppressor of the pro-apoptotic gene *TP53INP1.* In contrast, other relevant downregulated tumor-suppressor miRNAs, like miR141 and miR200c, have been described in in Mycosis Fungoides (MF) tumor stage as members of the Notch pathway, activated by upregulation of Jagged1 [301,302].

Epigenetic changes, like methylation, demethylation and acetylation, may act directly on chemical transformations of DNA and related DNA proteins, as happens in histones [303]. EZH2 mediates the histone H3 lysine 27 (H3K27) trimethylation and has been identified highly upregulated in patients with cutaneous ALCL. Furthermore, EZH2 promotes disease progression through histone methyltransferase activity in pcALCL [304]. Silencing T-cell transcription factors such as *GATA3*, *LEF1* and *TCF1* has been shown to occur by H3K27 trimethylation in ALCL cells [305].

### 6.6. Molecular Biomarkers and Prognosis

The prognosis of primary cutaneous CD30+ lymphoproliferative disorders is favorable in most cases and, indeed, LyP and pcALCL patients have an outstanding prognosis [306]. Recent comprehensive data on pediatric LyP also revealed a good prognosis and excellent survival rates [307]. Although pcALCL is rare in children, the prognosis is also typically good, like LyP, and is associated with a favorable clinical course. Nevertheless, continued close monitoring is recommended [308].

The prognosis of pcALCL does not change when regional lymph nodes are affected, but the disease behaves like systemic ALCL ALK- when it affects non-regional lymph nodes from other areas. The reason for this is still not fully understood, and further research is needed to determine how it differs from systemic manifestations that infiltrate the skin [309]. No large studies exploring molecular features have compared systemic ALCL affecting the skin with pcALCL that subsequently becomes systemic. On the other hand, the morphology of pcALCL with a *DUSP22* gene translocation differs from that of systemic ALCL that infiltrates the skin and presents the same translocation [180].

Theoretically, morphology does not imply a prognosis, but when considering the clinical data, it raises the differential diagnosis with other entities such as MF, epidermotropic CD8, gamma-delta lymphoma or NK/T-Cell Lymphoma, which generally follow a more aggressive clinical course [310].

### 6.7. Differential Diagnosis of pcALCL with Other CD30+ Cutaneous Entities

The histological overlap between LyP subtype C and pcALCL makes it a challenge to differentiate the two entities based on morphology. They are also clinically different: pcALCL generally forms single lesions that may self-resolve, while LyP usually debuts with multiple affected regions that appear and disappear over time. The *TCR* rearrangement is much more frequently detected than type A and type B LyP [289]. A multicenter study demonstrated that MF and pcALCL are hematological malignancies most associated with LyP [311].

A difficult differential diagnosis to perform is that distinguishing the papular variant of MF and LyP type B. Although the MF variant may be CD30-negative, and LyP type B variant more weakly expresses CD30 immunoreactivity, histopathologically it mimics LyP type B [312].

The LyP type B variant should also be compared with MF. It is CD4+ and has CD30+ small intraepidermal cells. There is also a variety of papillomatous MF, type C, that has an aggressive epidermotropic phenotype. The type E should be assessed towards natural killer T-Cell Lymphoma which is not EBER positive. All variants against pityriasis lichenoides et varioliformis acuta (PLEVA), especially the type D variant, should be considered. There is also a variety of gamma-delta LyP, which should be evaluated in relation to gamma-delta T-Cell Lymphoma [313].

*ALK* rearrangement is useful for differentiating these two entities because this genomic alteration is very rarely reported in pcALCL [314]. Further positive immunohistochemistry for EMA suggests secondary skin association in systemic ALCL rather than in pcALCL [280,315]. In contrast, pcALCL may express cutaneous lymphocytic antigen and CCR4, which are not seen in secondary cutaneous ALCL [316].

MF with CD30+ large cell transformation (CD30+ TMF) is another entity worth considering [317]. Expression of GATA3 appears to be a useful marker, with its strong and diffuse immunohistochemical pattern that is negative or weakly positive in pcALCL [318]. However, immunostaining studies appear to show some disparity in GATA3 expression with a lack of sensitivity and specificity [318]. Conversely, it seems that galectin-3 has a positive pattern of expression in pcALCL and shows a much lower level of expression in MF with CD30+ large cell transformation [319]. *CDKN2A/2B* deletions were also found in patients with CD30+ TMF in greater proportions than in CD30+ T-cell lymphoproliferative disorders [297]. Differentiating pcALCL from CD30+ TMF is challenging. Novel whole-transcriptome comparison and screening studies have been performed, the valuable findings revealing distinct clinicopathological features and unique gene expression markers. CD30+ TMF showed enrichment of T-cell receptor signaling pathways and an exhausted T-cell phenotype, whereas cALCL cells expressed high levels of HLA class II genes, orientated towards a Th17 phenotype, as well as neutrophil infiltration. Additional immunohistochemical algorithms have even been proposed to distinguish the entities: BATF3+ and TCF7- expression indicates a cALCL pattern, and a TCF7+ and BATF3- phenotype, which defines a CD30+ TMF pattern [320].

*DUSP22* translocations are seen in pcALCL and LYP, and rarely in cases of MF. This expands the spectrum of *DUSP22*-rearranged lymphomas to incorporate MF-like presentations that do not show large cell transformation [321]. There are other case reports of sequential presentations of pcALCL, MF-like and LyP-like lesions, each with a *DUSP22* translocation [322]. A recent case report demonstrated the detection of a *DUSP22* translocation and a common clonal T-cell receptor rearrangement in a patient with histological MF-like and pcALCL features [323]. However, while diagnosis of pcALCL requires the exclusion of clinical evidence of MF, multiple cases with co-existing MF with pcALCL and/or LyP have been reported, which complicates the distinction between the entities [324]. *DUSP22* rearrangement was also described in a rare case of cutaneous presentation of enteropathy-associated T-cell lymphoma by Bisig et al. [325].

Distinguishing between tumor-stage MF and conditions expressing CD30+ such as ALCL is difficult due to their overlapping features. It requires clinical evaluation, and histological and immunophenotypical examination [281]. A recent study provided evidence about the positive expression of the *TOX* gene in advanced-stage MF [326], whereas its expression is not so elevated in Cutaneous T-Cell Lymphoma [327,328]. Tumor-stage MF is usually characterized by small-to-medium-sized pleomorphic cells and B-cells, lymphoid follicles, Langerhans cells and eosinophils may also be present [328].

Differential diagnosis with B-cell lymphomas with CD30+ large cells in immunosuppressed patients should be made. This entity is positive for the expression of Epstein–Barr Virus (EBV), which is absent from pcALCL [74]. This distinction should be also made with B-cell lymphoma with plasmablastic differentiation involving the skin, which is usually positive for EBV, ruling out the possibility of pcALCL [329]. No cytotoxic markers in plasmablastic differentiation and expression of MUM1 can be found in both entities [330]. Primary effusion lymphoma, an unusual form of aggressive B-cell lymphoma associated with herpesvirus 8 [331], is another entity involving the skin that merits consideration.

Diagnosis of pcALCL with other non-tumoral CD30+ entities include reactive/inflammatory cutaneous conditions like pyoderma, sweet syndrome and fungal infections where scarce immunostained CD30+ immunoblasts are seen instead of the strong and diffuse expression of CD30 [281,332].

## 7. Microenvironment in ALCL

Composition and crosstalk between the neoplastic cells and tumor microenvironment of ALCL needs further investigation. Different studies characterized molecules that contribute to modify ALCL tumor microenvironment, but the bibliography about this topic is scarce. Future studies, particularly those integrating spatial transcriptomics and multiplexed immunohistochemistry will be essential to map immune subsets in the ALCL microenvironment, clarifying how they influence response or resistance to emerging immunotherapies in ALCL. Some studies discovered potential therapeutic targets in ALCL due to their role in the tumoral microenvironment, as well as the role of immune subpopulations in tumor progression.

### 7.1. ALK+ ALCL

Exosomes polarizing fibroblast from bone marrow to cancer-associated fibroblast (CAF) in ALK+ ALCL have been described, altering the cytokine profile of the microenvironment, contributing to tumor aggressiveness and resistance to treatment [333]. Another relevant immune subpopulation in ALCL are tumor-associated macrophages (TAMs), with differences between ALK+ and ALK- ALCLs. ALK+ ALCL had a higher expression of PD-L1 in the tumor cells, in contrast to ALK- ALCL, which expressed high PD-L1 in TAMs [334].

Some studies revealed key players in ALK+ ALCL microenvironment modulation, such as RNY4. This protein has a major role modulating the tumor microenvironment, and fragments loaded into exosomes of pediatric ALK+ ALCL patients correlated with more advanced and aggressive disease [335]. Another finding is related with IL-2 protein expression, which generally appears in background cells of the microenvironment and not in tumoral cells. ERK1/2 signaling activated by this cytokine enhances lymphoma cell survival in vitro, highlighting its importance [336]. Finally, miR-135b mediated oncogenicity in ALK+ ALCL has been described. This short transcript increases IL-17 release, modifying the immunophenotype of tumoral cells, which “mimic” Th17 cells to escape immune system [123].

DNA nanomicelles loaded with doxorubicine and *ALK*-specific siRNA induced apoptosis of ALCL K299 cells in vitro and inhibited tumor growth in vivo, potentially representing another therapeutical approach to treat these tumors [337].

### 7.2. ALK- ALCL

The study of Drieux et al. [338] segregated 34 ALK- ALCL cases into two clusters. The first one comprised a cytotoxic cluster (Th1-polarized) with cases overexpressing *PRF* and *GZMB* (n = 10). The second, non-cytotoxic cluster (Th2 signature) was enriched in DUSP22R cases that showed upregulation of Th2 markers like *GATA3* and *CCR4* (n = 24). This study proposes a simple gene expression signature to classify PTCLs in general. The association between the presence of DUSP22R and the Th2 marker signature was remarkable, and could be applied to routinely fixed samples for diagnostic procedures.

Th1-polarized tumors are thought to have increased immunogenicity and potential sensitivity to checkpoint inhibitors. Conversely, Th2-rich ALCLs may show an immunosuppressive microenvironment leading to resistance against PD-1/PD-L1 blockade-based therapies [92,175]. This Th2-rich profile is remarkable in DUSP22R, characterized by a “cold” microenvironment (less inflammatory and citotoxic) due to their low expression levels of PD-L1, suggesting limited efficacy of PD-1-targeting therapies [175,339,340]. In contrast, the rest of ALK- ALCLs demonstrate robust PD-L1 expression pointing to their potential as candidates for immune checkpoint inhibitors [210]. Nevertheless, PD-L1 expression did not appear to correlate with clinical outcomes between the different ALCL subgroups [210,341].

Finally, Gal-1 expression correlates with adverse outcome in ALK- ALCL patients, which show less cytotoxic T cells in their microenvironment compared to Gal-1 depleted cases [342].

### 7.3. BIA-ALCL

Inflammatory microenvironment appeared to be crucial for BIA-ALCL neoplastic cells malignant transformation due to its role activating JAK/STAT3 pathway [343]. Another key finding is that Mesenchymal Stem Cells (MSCs) biology may be modified by neoplastic ALCL cells as well. In pro-inflammatory ALCLs, MSCs release a different immunoregulatory cytokine profile which favor host immune evasion [344].

### 7.4. pcALCL

In pcALCL, reactive inflammatory infiltrate is also important for development of tumoral niche as well, highlighting the substantial proportion of eosinophilic infiltration observed in these patients [279,345]. Differences in M2 CD163+ TAMs between pcALCL and CD30-rich reactive lymphocytes skin has been observed as well [346]. These differences regarding TAMs presence may explain their different clinical behavior and prognosis, but the major overall presence of TAMs is related to worse clinical outcomes [347].

EZH2 functions as an inhibitor of CD4+ and CD8+ effector T-cell recruitment into the tumor microenvironment in pcALCL, favoring immune evasion, tumor progression and antitumor immunity, making it an interesting pharmacological target [304].

## 8. Future Direction of ALCL Diagnosis

Molecular advances in ALCL diagnosis in recent years have significantly changed our understanding of the biology of these tumors and enabled considerable progress in their classification to be made. However, many questions remain to be answered, and further research is required, especially to clarify the effects of the various rearrangements and alterations described in this review. We expect a paradigm shift to occur in the near future in the classification of ALCLs, due in particular to the development and implementation of advanced molecular techniques in routine procedures that help us optimize the diagnoses of this remarkably heterogeneous class of lymphomas.

Over the last decade, many algorithms and molecular signatures have been proposed that would be suitable for diagnostic purposes. These efforts have demonstrated that there are many aspects of the behavior of these lymphomas that directly affect their classification in the clinical milieu, but about which we are currently ignorant. The unification of some of these signatures and their inclusion in multicenter studies for the purpose of checking the viability of these algorithms is recommended since it is likely to advance our knowledge about this pathology.

NGS is now an essential clinical tool, enabling the detection of rare oncogenic events. For example, in a recent case NGS revealed a *TRAF1::ALK* fusion in a patient initially diagnosed with a poorly differentiated sarcoma; this prompted a revised diagnosis of stage IV ALK-positive ALCL and led to brentuximab vedotin-based therapy with lasting remission beyond 28 months [348]. Targeted NGS panels for T-cell receptor clonality have also improved sensitivity in FFPE samples, surpassing conventional PCR methods and offering potential for more accurate minimal residual disease monitoring.

The emergence of spatial transcriptomics is remarkable as well. This technology allows to integrate gene expression data with histological architecture. Although still nascent in ALCL, spatial transcriptomics in other lymphomas has mapped critical tumor–immune cell interactions, highlighted microenvironmental heterogeneity, and uncovered zones of immunosuppression with prognostic relevance [349].

The integration of these techniques through multi-omics is particularly promising. By combining genomic, transcriptomic, and spatial data, researchers are beginning to build predictive biomarker models which could be included and improve the precision of the diagnoses.

Together, these approaches deliver key benefits, such as discovery of rare mutations or fusions invisible to IHC or FISH, spatially resolved insights into subclonal architecture and immune microenvironment and data-driven biomarker models to support targeted therapy decisions. Integrating these novel technological approaches into routine diagnostics represents the next step “beyond morphology and immunophenotype”, establishing a robust precision-pathology paradigm capable of guiding personalized treatment strategies, which is encouragely needed to improve the management of these patients.

## Figures and Tables

**Figure 1 ijms-26-05871-f001:**
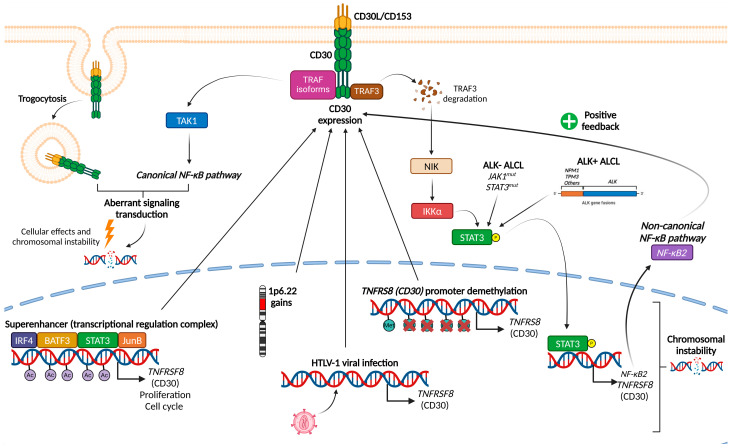
Different mechanisms leading to CD30 overexpression. Created with BioRender.com.

**Figure 2 ijms-26-05871-f002:**
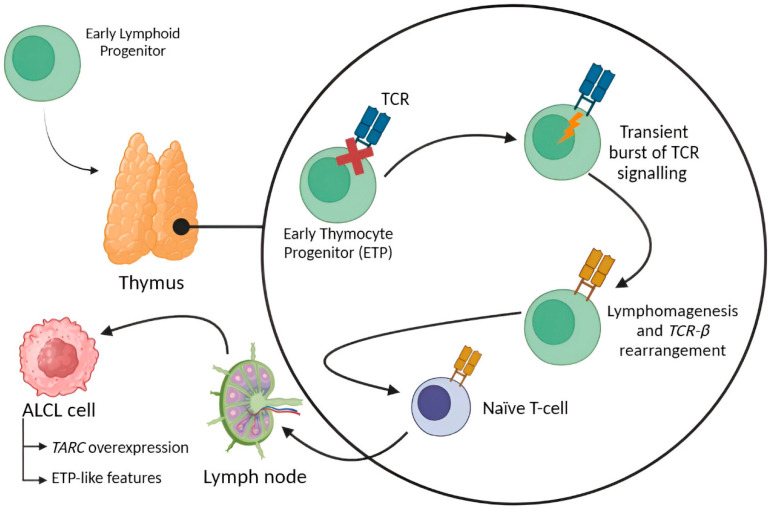
ALCL etiopathogenesis most accepted theory is supported by TARC overexpression, as well as the presence of ETP-like characteristics. Created with BioRender.com.

**Figure 3 ijms-26-05871-f003:**
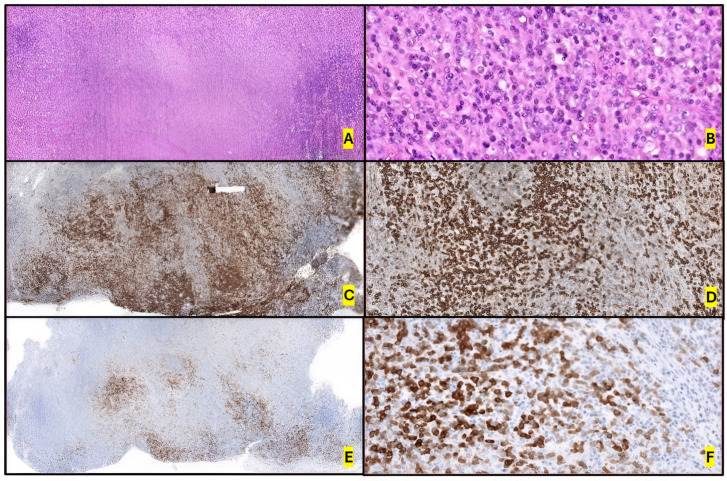
Neoplastic small-to-medium-sized cells with homogeneous appearance are partially disrupting lymph node architecture ((**A**), HE, 10× and (**B**), HE, 40×). Neoplastic cells express CD30 ((**C**), 10× and (**D**), 20×) and ALK with a peculiar exclusive cytoplasmic pattern ((**E**), 10× and (**F**), 40×).

**Figure 4 ijms-26-05871-f004:**
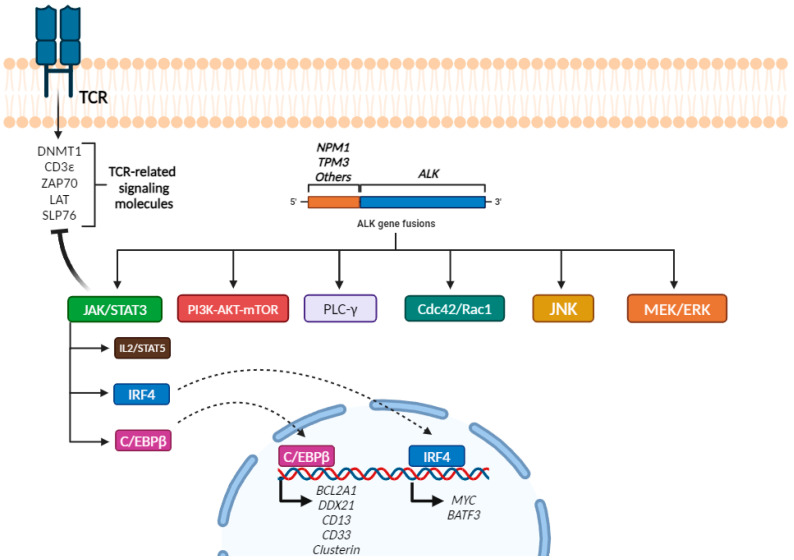
Signaling profile of ALK+ ALCL. Created with BioRender.com.

**Figure 5 ijms-26-05871-f005:**
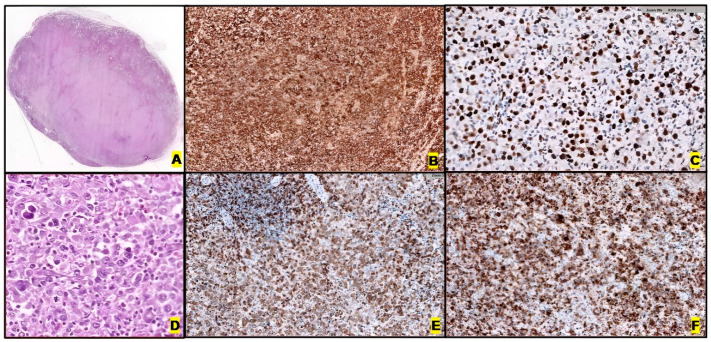
A triple-negative ALCL case for ALK, P63 and DUSP22 translocation is shown. (**A**) Effacement of the lymph node by a pseudonodular pattern of large atypical cells is seen (HE, 10×). (**D**) Large atypical cells with different nuclear morphology and size is shown (HE,40×). Neoplastic cells expressed CD30 ((**B**), 20×), intense nuclear PSTAT3 ((**C**), 40×), cytoplasmic CD3 ((**E**), 20×) and perforin ((**F**), 20×).

**Figure 6 ijms-26-05871-f006:**
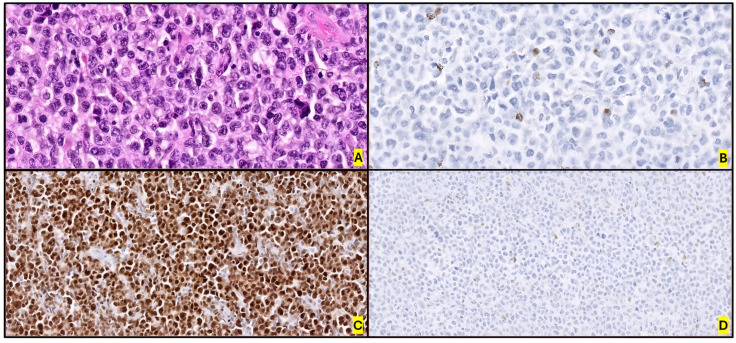
A typical case of DUSP22-rearranged ALCL. Numerous “hallmark cells” with “donut-like” nuclear figures are easily found ((**A**), HE, 40×). Neoplastic cells were pSTAT3 negative ((**B**), 40×), LEF1 positive ((**C**), 40×) and TIA1 negative ((**D**), 20×).

**Figure 7 ijms-26-05871-f007:**
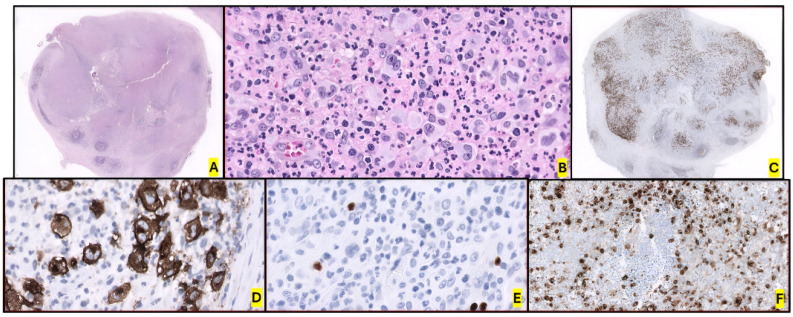
Example of an ALK- ALCL case with JAK2 translocation is shown. (**A**) Partial effacement of a lymph node by different-sized nodules separated from each others by collagen bundles and inflammatory elements (HE, 4×). (**B**) At higher-power view, large atypical cells with Hodgkin’s-like looking features intermingled with leukocytes, eosinophils and small lymphocytes are found (HE, 40×). (**C**) Neoplastic cells express CD30 (4×). (**D**) Higher-power view of CD30 expression (40×). (**E**) No PAX5 is expressed on neoplastic cells (20×). (**F**) Expression of perforin is seen (10×).

**Figure 8 ijms-26-05871-f008:**
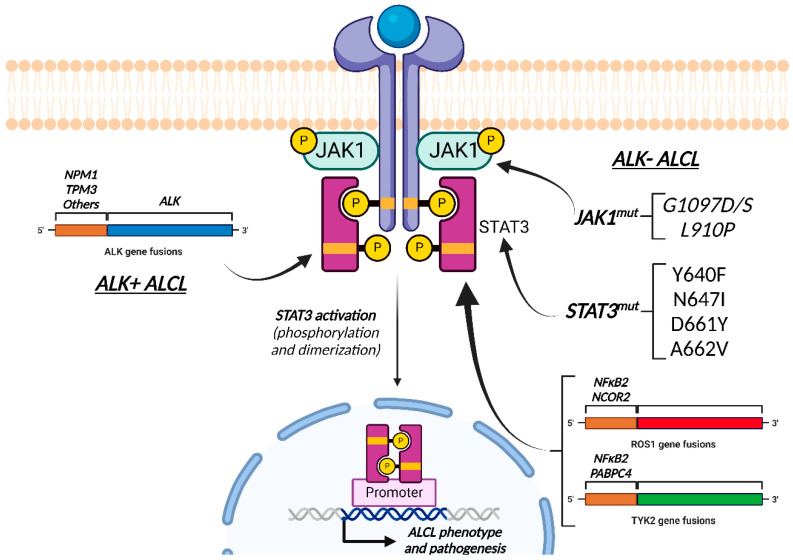
Comparison between the different mechanisms for STAT3 activation between ALK+ and ALK- ALCL. Created with BioRender.com.

**Figure 9 ijms-26-05871-f009:**
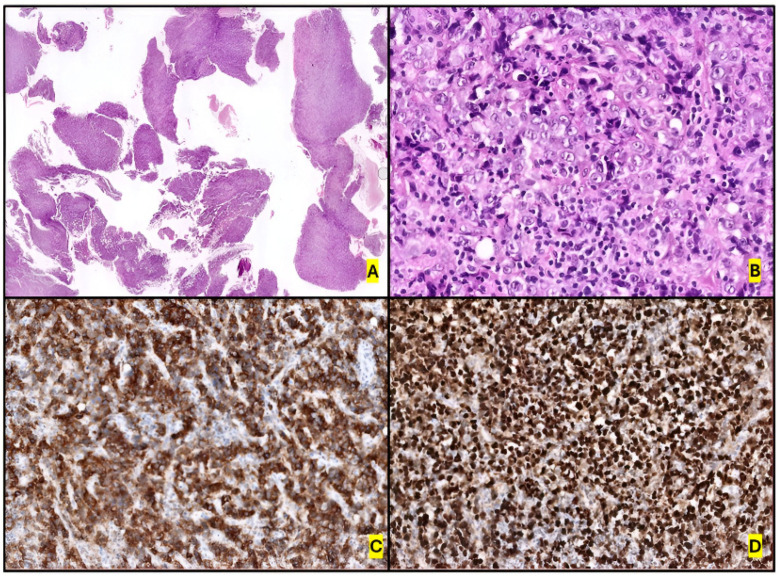
Example of an ALK- ALCL with p53 overexpression. (**A**) Complete effacement of lymph node architecture by a diffuse proliferation of large atypical cells (HE staining, 10×). (**B**) Higher magnification 20×. (**C**) CD30 expression (20×). (**D**) Intense p53 expression (20×).

**Figure 10 ijms-26-05871-f010:**
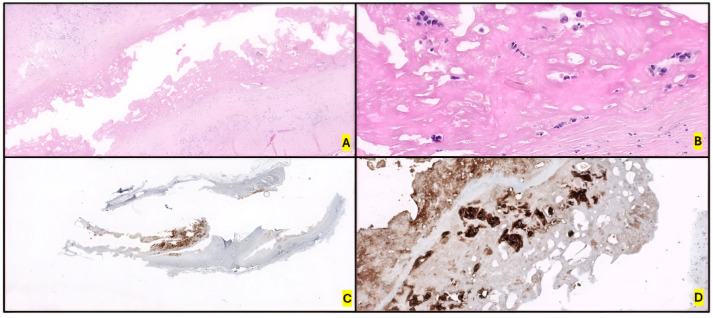
Example of a BIA-ALCL. (**A**) A breast protesis capsule with fibrinous material in its lumen is shown (HE, 4×). (**B**) Higher-power view of the fibronous areas show large atypical cells alone or forming small clusters (HE, 40×). (**C**) Less magnification (2×) and (**D**) higher magnification (40×) show CD30 expression on neoplastic cells.

**Figure 11 ijms-26-05871-f011:**
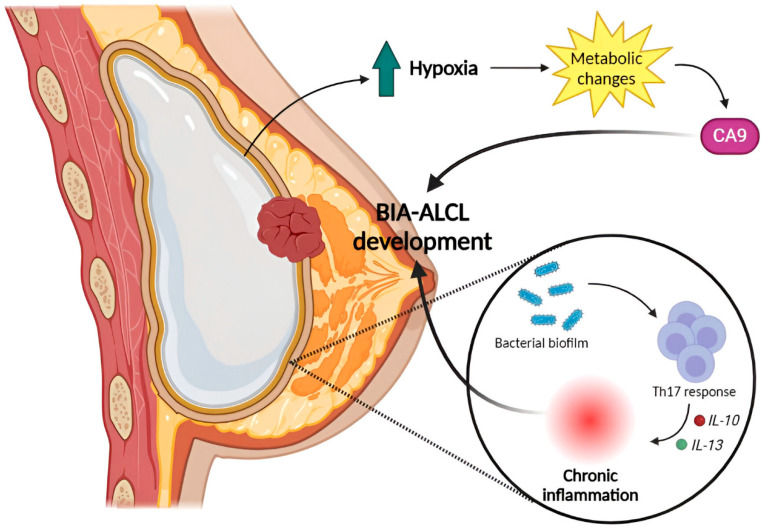
Representation of the two main theories established for the development of BIA-ALCL (hypoxia-related and chronic inflammation due to the presence of a bacterial biofilm along the implant). Created with BioRender.com.

**Figure 12 ijms-26-05871-f012:**
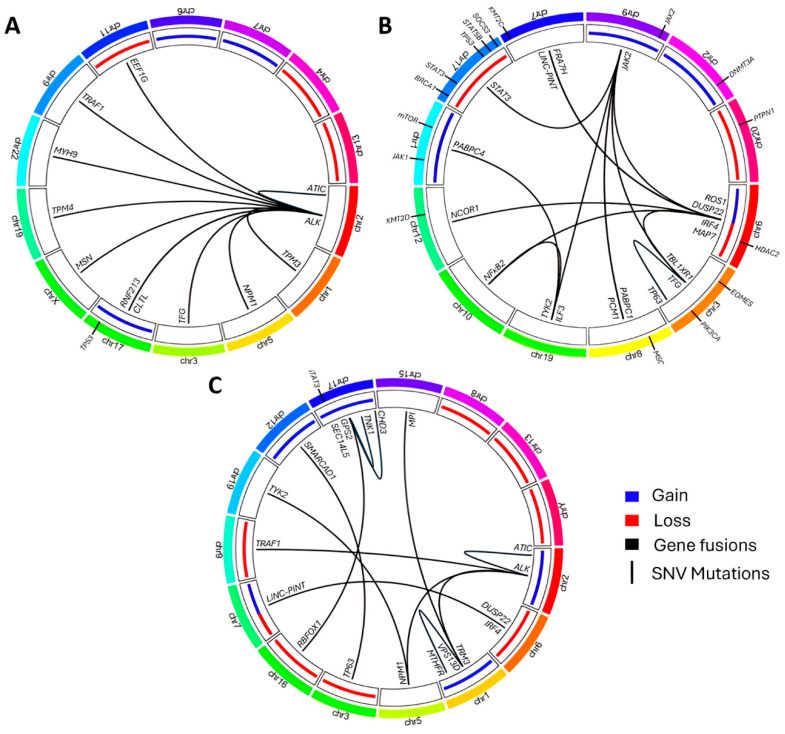
Circoplots showing the diverse landscape of gene rearrangements, chromosomal alterations (gains or losses) and SNV mutations observed in different subsets of ALCL. (**A**) ALK+ ALCL, (**B**) ALK- ALCL (including BIA-ALCL) and (**C**) pcALCL.

**Table 1 ijms-26-05871-t001:** CD30 expression in different T-cell hematological malignancies. PTCL-NOS: Peripheral T-Cell Lymphoma, Not Otherwise Specified. CTCL: Cutaneous T-Cell Lymphoma. MF: Mycosis Fungoides. AITL: Angioimmunoblastic T-Cell Lymphoma. ATLL: Adult T-cell Leukemia/Lymphoma. ENKTL: extranodal NK/T-Cell Lymphoma. EATL: enteropathy-associated T-Cell Lymphoma. F-PTCL: Follicular Peripheral T-Cell Lymphoma. NA: not available.

Study Reference	CD30+ Cutoff	PTCL-NOS	CTCL/MF	ALK- ALCL	ALK+ ALCL	AITL	ATLL	ENKTL	EATL	F-PTCL
Sabattini et al., 2013 [21] (N = 192)	0: No staining 1: 0 to <25% 2: 25 to 50% 3: >50 to 75% 4: >75%	36% 13% 21% 13% 18%	41% 47% 6% 0% 6%	NA	NA	51% 21% 12% 10% 0%	NA	20% 10% 30% 10% 30%	0% 0% 22% 0% 78%	NA
Bossard et al., 2014 [22] (N = 376)	0: No staining 1: 0 to <25% 2: 25 to 50% 3: >50 to 75% 4: >75%	42% 26% 9% 10% 13%	NA	0% 0% 0% 0% 100%	0% 0% 5% 2% 93%	37% 47% 10% 5% 0%	44% 11% 33% 11% 0%	54% 7% 11% 14% 14%	50% 0% 0% 7% 43%	NA
Lamarque et al., 2016 [23] (N = 56)	0: <5% 1: 5 to 24% 2: 25 to 49% 3: 50–75% 4: >75%	10% 10% 30% 30% 20%	14% 0% 0% 14% 71%	0% 0% 0% 0% 100%	0 0 20 20 60	0% 100% 0 0 0	100% 0% 0% 0% 0%	NA	0% 100% 0% 0% 0%	NA
Rodriguez-Pinilla et al., 2021 [24] (N = 175)	Mean (SD) of % stained cells in each group	25.0% (34.8)	NA	74.5% (32)	97.3% (6.5)	18.5% (21)	NA	53.1% (41)	31.7% (46.6)	NA
Karube et al., 2008 [25] (N = 319) *	>70% 20 to 70%	5% 11%	9% 9%	58% 35%	NA	0% 32%	15% 24%	0% 64%	NA	NA
Asano et al., 2011 [26] (N = 47)	>30%	51%	NA	NA	NA	NA	NA	NA	NA	NA
Savage et al., 2008 [27] (N = 490)	>0% ≥80%	32% 5%	NA	100% 100%	100% 100%	NA	NA	NA	NA	NA
Weisenburger et al., 2011 [28] (N = 217)	>20%	32%	NA	NA	NA	NA	NA	NA	NA	NA
Wang et al., 2017 [29] (N = 122)	0: No staining 1: 0 to <25% 2: 25 to 50% 3: >50 to 75% 4: >75%	NA	NA	NA	NA	NA	NA	30% 38% 18% 10% 5%	NA	NA
Kawamoto et al., 2018 [30] (N = 97)	0: ≥1% 1: ≥10% 2: ≥20%	NA	NA	NA	NA	NA	NA	57% 55% 44%	NA	NA
Feng et al., 2017 [31] (N = 622)	≥20%	NA	NA	NA	NA	NA	NA	47.3%	NA	NA
Hartmann et al., 2019 [32] (N = 16)	>60%	NA	NA	NA	NA	NA	NA	NA	NA	75%
Onaindia et al., 2016 [33] (N = 51)	0: No staining 1: 0 to <25% 2: 25 to 50% 3: >50	25% 0% 37.5% 37.5%	NA	NA	NA	7% 83.7% 9.3% 0%	NA	NA	NA	NA
Shen et al., 2024 [34] (N = 82)	1: <40% 2: ≥40%	NA	NA	NA	NA	NA	NA	18.3% 81.7%	NA	NA

* Study did not distinguish between ALK+ and ALK- ALCL.

**Table 3 ijms-26-05871-t003:** Summary of the non-coding RNAs described in ALK+ ALCL.

Non-Coding RNA Type	Transcript	Regulation Status	Effect
miRNA	miR-17-92	Overexpressed	Activation of STAT3
	miR-101	Downregulated	Activation of mTOR pathway
	miR-155	Downregulated	Create immunosuppressive microenvironment, favoring Th2 differentiation due to low levels of IFN-γ
	miR135b	Overexpressed	Activation of NPM–ALK–STAT3 axis. Favor immunosuppressive microenvironment due to IL-17 secretion (Th17 signature), activating the transcription of *STAT6* and *GATA3*
	miR-150	Downregulated	Protumoral properties
	miR-146a-5p	Overexpressed	M2 macrophage infiltration, improving tumor aggressiveness and dissemination
lncRNA	LINC01013	Overexpressed	Induction of Snai1 (activating EMT)
snoRNA	All	Downregulated	Unknown
	U3	Overexpressed	Unknown, but serves as tool for diagnostic procedures
circRNA	All	Overexpressed	Unknown. Formation of these non-coding RNAs between the breakpoint of *NPM* and *ALK*

**Table 4 ijms-26-05871-t004:** New proposal for ALK- ALCL molecular subtypes classification and their main features. +: Positive. -: Negative. NA: not available.

	DUSP22R	TP63R	TN pSTAT3+	TN pSTAT3-	“Double Hit” DUSP22R and TP63R
Neoplastic cells	Hallmark cells Monomorphism Doughnut cells	Pleomorphic cells Mitosis	Sheet-like neoplastic cells, large pleomorphic cells	Pleomorphic	NA (absence of DUSP22R morphology)
Inflammatory background	Scattered	>lymphocytes	Lymphocyte-rich background	Monomorphic	NA (abscence of DUSP22R morphology)
IHC	Non-cytotoxic phenotype (TIA1 < 20%) P63 variable LEF1 > 75%	Cytotoxic phenotype P63 > 30%	Cytotoxic phenotype	Non-cytotoxic phenotype	Non-cytotoxic phenotype
DUSP22R	+	-	-	-	+
TP63R	-	+	-	-	+
JAK/STAT3 pathway activation	-	+	+	-	NA
pSTAT3 expression	-	+	+	-	NA
*TP53* mutations	-	+/-	-/+	+/-	NA
5-year OS	40–90%	20%	50%	20%	NA

**Table 5 ijms-26-05871-t005:** Main differences between ALK- ALCL and CD30+ PTCL-NOS.

Characteristic	ALK- ALCL	CD30+ PTCL-NOS
Immunophenotype	Strong CD30, EMA+; CD3 may be absent or weak	CD30 variable, CD3 strong, other T markers present (CD4, CD8)
DUSP22 rearrangement	Common (~30% of cases)	Rare
TP63 rearrangement	Less common (~8% of cases)	Rare
TNFRSF8 (CD30), BATF3, TMOD3 gene expression	Highly expressed; differentiates with 97% accuracy	Less common expression
Loss of 5q/9p	Rare	Common
STAT3 phosphorylation (pSTAT3-S727)	Overexpressed; strong marker for ALK- ALCL	Rarely overexpressed
Cytotoxic phenotype markers (TIA-1, granzyme B)	Frequently positive	Infrequent or absent
Epigenetic mutations (e.g., TET2, DNMT3A)	Rare	Frequently mutated

**Table 6 ijms-26-05871-t006:** Main molecular alterations described in BIA-ALCL.

Recurrent Molecular Alterations	Frequency (%)
**JAK/STAT signaling**	
STAT3	11–64
JAK1	7–44
SOCS1	3–20
SOCS3	6
PTPN1	3–9
**Epigenetic modifiers**	
KMT2C	11–26
CHD2	15
CREBBP	15
KMT2D	9
DNMT3A	6–20
HDAC2	6
TET2	3
**Cell cycle/apoptosis**	
TP53	11–20
**Chromosomal alterations**	
Loss of 20q13.13	66
Focal amplification of 9p24.1	33

**Table 7 ijms-26-05871-t007:** Immunophenotype summary of the main ALCL entities. +: positive. -: negative. +/-: variable. +++: high positivity.

Marker	ALK+ ALCL	ALK- ALCL: *DUSP22*-R	ALK- ALCL: *TP63*-R	ALK- ALCL: Triple Negative	BIA-ALCL	Primary Cutaneous ALCL
Lymphoid origin						
CD45	-	+	+	+	+	+
CD3	25%	+/-	+/-	+/-	+/-	+/-
CD4	40%	70%	+	+	+	+
CD8	-	-	-	-	-	+/-
CD2	30%	+/-	+	+	+	+
CD5	30%	+/-	+/-	+/-	-	+/-
CD7	+/-	+/-	+/-	+/-	+/-	+/-
ALK (Anaplastic Lymphoma Kinase)	+	-	-	-	-	-
Cytotoxic markers (TIA1, granzyme B, and perforin)	75–90%	-	+	+/-	+	+
CD30	+++	+++	+++	+++	+++	+++
CD20	-	-	-	-	-	-
EBER	-	-	-	-	-	-
Epithelial origin (EMA and/or Cytokeratins)	30%	-	-	-	Only EMA in excepctional cases	-
Ki-67	High (>70%)	Moderate–High (50–70%)	High (>80%)	Variable (40–70%)	High (>70%)	Variable (40–70%)
Others	CD43 (30%)	LEF1 (>75%)	P63 (35%)		CD43	LEF1 and PSTAT3 in *DUSP22*-R cases
	CD56 (10%)	TCRαβ (<20%)	pSTAT3 (rarely)		CD25	Negative for cytotoxic markers in DUSP22-R cases
	pSTAT3 (75%)	pSTAT3 (<20%)			TCRαβ or TCRγδ (30%)	PD-1 and/or ICOS (rarely)
	CD15 (rarely)	PAX5 (rarely)				CD15 (45%)
	PAX5 (rarely)	CD138 (rarely)				CD56 (rarely)
		BCL2 negative				TCRγδ (rarely)

**Table 8 ijms-26-05871-t008:** Summary of the main molecular abnormalities described in each main type of ALCL.

Entity	Abnormality Type	List of Alterations
ALK+ ALCL	Chromosomal translocation (ALK partner)	t(2;5)(p23;q35) (*NPM1*), t(1;2)(q25;p23) (*TPM3*), Inv(2)(p23q53) (*ATIC*), t(2;3)(p23;q21) (*TFG*), t(2;17)(p23;q23) (*CLTL*), t(2;X)(p23;q11.12) (*MSN*), t(2;19)(p23;p13.1) (*TPM4*), t(2;22)(p23;q11.2) (MYH9), t(2;9)(p23;q33–34) (*TRAF1*), t(2;11)(2p23;11q12.3) (*EEF1G*), t(2;17)(p23;q25) (*RNF213/ALO17*)
	Chromosomal alterations	6q (gain), 7p (gain), 17p/17q24 (gain), 4q13-q21/11q14 (loss), 13q (loss)
	Mutations	*TP53*
ALK- ALCL	Gene fusions and chromosomal translocations	*DUSP22::FRA7H* t(6;7), *DUSP22/IRF4::LINC-PINT* t(6;7), *TBL1XR1::TP63* inv(3)(q26q28), *DUSP22::TBL1XR1* t(6;3), *PABPC1::JAK2* t(8;9), *TFG::JAK2* t(3;9), *ILF3::JAK2* t(19;9), *MAP7::JAK2* t(6;9), *PCM1::JAK2* t(8;9), *NFκB2::ROS1* t(10;6), *NCOR2::ROS1* t(12;6), *NFκB2::TYK2* t(10;19), *PABPC4::TYK2* t(1;19), *FRK::PABPC1* t(6;8), *FRK::MAPK9* t(6;5), *FRK::CAPRIN1* t(6;11), *MYC::Unknown* t(8;?)
	Chromosomal alterations	1q (gain), 6p21 (gain), 6q21 (loss), 17p13 (loss), 2 Trisomy
	Mutations	*JAK1^G1097D/S^*, *JAK1^L910P^*, *STAT3^Y640F^*, *STAT3^N647I^*, *STAT3^D661Y^*, *STAT3^A662V^*, *TP53*, *MSC^E116K^*, *PRDM1*, *BANK1*, *FAS*, *STIM2*, *LRP1B*, *EPHA5*, *KMT2D*
BIA-ALCL	Gene fusions	STAT3::JAK2 t(17;9)
	Chromosomal alterations	9p24.1 (gain), 20q13.12–13.2 (loss)
	Mutations	*TP53*, *BRCA1/2*, *JAK1*, *JAK2*, *STAT3*, *STAT5B*, *SOCS1*, *SOCS3*, *PTPN1*, *KMT2C*, *KMT2D*, *CHD2*, *CREBBP*, *DNMT3A*, *TET2*, *HDAC2*, *EOMES*, *PI3K/AKT/mTOR*
pcALCL	Gene fusions and chromosomal translocations	*ALK::NPM1* t(2;5), *DUSP22/IRF4::LINC-PINT* t(6;7), *ALK::TRM3* t(1;2), *ALK::ATIC* t(2;2), *ALK::TRAF1* t(2;9), *NPM1::TYK2* t(5;19), *TP63::SMARCAD1* t(3;12), *GPS2::TNK1* inv(17), *GPS2::CHD3* inv(17), *RBFOX1::SEC14L5* t(16;17), *VPS13D::MPI* t(1;15), *VPS13D::MTHFR* inv(1)
	Chromosomal alterations	Gain of 1p/q, 2p/q, 7q31, 12q, 17q and loss of 3p, 6q, 7p/q, 8p, 9p21, 13q, 16q, Y
	Mutations	*STAT3*

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
