# Peer review of "Molecular Insights into the Diagnosis of Anaplastic Large Cell Lymphoma: Beyond Morphology and Immunophenotype"

_ijms, 2025, doi:10.3390/ijms26125871_

Round 1
Reviewer 1 Report
Comments and Suggestions for Authors
In this review, the authors comprehensively summarize the molecular pathological findings of anaplastic large cell lymphoma (ALCL). This review is highly informative and valuable for understanding current research on ALCL. Below are several suggestions that may help improve the manuscript:
- The title includes the phrase “beyond morphology and immunophenotype”, but the content of the review mainly discusses molecular pathology in the context of conventional pathology-based diagnosis. Therefore, the use of “beyond” may be somewhat of an overstatement. To better match the title, it may be beneficial to add a final paragraph providing a perspective on the future direction of ALCL diagnosis.
- The inclusion of actual histological images is very helpful for understanding the pathological findings. However, it is unclear how these images are cited or sourced. If these are original images, it should be clearly stated. In relation to this point, lines 336–337 mention EZH2 staining, but the histological images do not appear to show the EZH2 staining results.
- Some of the content in Figures 8 and 12 appears to overlap. Combining these figures may help to present the information more clearly and in a more organized manner.
Author Response
Reviewer #1
In this review, the authors comprehensively summarize the molecular pathological findings of anaplastic large cell lymphoma (ALCL). This review is highly informative and valuable for understanding current research on ALCL. Below are several suggestions that may help improve the manuscript:
The title includes the phrase “beyond morphology and immunophenotype”, but the content of the review mainly discusses molecular pathology in the context of conventional pathology-based diagnosis. Therefore, the use of “beyond” may be somewhat of an overstatement. To better match the title, it may be beneficial to add a final paragraph providing a perspective on the future direction of ALCL diagnosis.
Answer: We thank the reviewer for this comment. We agree that including a forward-looking perspective would strengthen the alignment between the title and the content of the review. To address this, we added new concluding paragraphs included in the last section of the manuscript, which name has been changed (8. Future direction in ALCL diagnosis). This addition highlights emerging technologies and integrative approaches that extend beyond current morphology- and immunophenotype-based classifications. This addition reinforces the concept of “beyond” in the title and contextualizes the discussed molecular findings.
- The inclusion of actual histological images is very helpful for understanding the pathological findings. However, it is unclear how these images are cited or sourced. If these are original images, it should be clearly stated. In relation to this point, lines 336–337 mention EZH2 staining, but the histological images do not appear to show the EZH2 staining results.
Answer: We appreciate this remark and we agree with it. We moved the “(Figure 5)” indication to a better location to not create confusion due to the lack of a EZH2 staining image in the panels included in Figure 5, although we maintained the text information because we consider is relevant for the manuscript. All images and figures are original, homemade and specifically designed for this review.
- Some of the content in Figures 8 and 12 appears to overlap. Combining these figures may help to present the information more clearly and in a more organized manner.
Answer: Following reviewers suggestions, we removed Figure 12 since all of it is written down in the text of the manuscript and we will keep Figure 8 since is quite visual.

Reviewer 2 Report
Comments and Suggestions for Authors
This is a well-constructed and comprehensive review. I support its acceptance, contingent upon minor revisions detailed below.
-
The authors should expand their discussion of the tumor immune microenvironment, particularly in the context of immunotherapy resistance. This is especially pertinent in the section addressing Th1/Th2 signatures (lines 458–464). The authors should comment on how these immune signatures might relate to or predict resistance to immunotherapeutic agents, particularly in light of emerging translational data.
-
The manuscript would benefit from a brief discussion of how recent advances in next-generation sequencing and spatial transcriptomics are being leveraged to identify and validate clinically actionable biomarkers in ALCL. These technologies are increasingly relevant for refining subtype classification, mapping tumor-immune interactions, and informing targeted therapy decisions. Their inclusion would underscore the evolving landscape of integrative diagnostics and further contextualize the review’s emphasis on precision medicine.
Author Response
This is a well-constructed and comprehensive review. I support its acceptance, contingent upon minor revisions detailed below.
- The authors should expand their discussion of the tumor immune microenvironment, particularly in the context of immunotherapy resistance. This is especially pertinent in the section addressing Th1/Th2 signatures (lines 458–464). The authors should comment on how these immune signatures might relate to or predict resistance to immunotherapeutic agents, particularly in light of emerging translational data.
Answer: We thank the reviewer for this comment. We agree that the immune microenvironment plays an important role in ALCL, and further research is needed due to the scarce bibliography about it. In response, we expanded the section, changed its name and location in order to highlight its importance (7. Microenvironment in ALCL). This new section of the manuscript includes a more detailed discussion about microenvironment in each ALCL subtype, explaining how different molecules and tumor microenvironment favor tumor immune evasion. We added recent studies exploring how these immune signatures might serve as predictors of response to different compounds. Our intention with the review was not to cover treatment details, but following reviewer´s comments, we added different studies describing novel therapeutical targets related with microenvironment in each section.
- The manuscript would benefit from a brief discussion of how recent advances in next-generation sequencing and spatial transcriptomics are being leveraged to identify and validate clinically actionable biomarkers in ALCL. These technologies are increasingly relevant for refining subtype classification, mapping tumor-immune interactions, and informing targeted therapy decisions. Their inclusion would underscore the evolving landscape of integrative diagnostics and further contextualize the review’s emphasis on precision medicine.
Answer: We appreciate this constructive suggestion, which is in line with what was proposed by the first reviewer. To incorporate the reviewer’s valuable input, we have expanded the final section of the manuscript (8. Future direction in ALCL diagnosis) to briefly discuss the emerging role of next-generation sequencing (NGS) and spatial transcriptomics in ALCL. These tools are indeed instrumental in identifying novel biomarkers, refining subclassification, and characterizing the tumor microenvironment in a spatially resolved manner. We hope that this addition is sufficiently informative to respond to your suggestion.
